# Research on state machine control optimization of double-stack fuel cell/super capacitor hybrid system

**Mengjie Li** [1], **Qianchao Liang**[1], **Jianfeng Zhao**[1], **Yongbao Liu**[1]*, **Yan Qin**[2]

**1** Department of Power Engineering, Naval University of Engineering, Wuhan, Hubei, China, **2** Dalian Naval Academy, Zhongshan District, Dalian City, Liaoning Province, China

* liuyongbaoly@163.com

**Data Availability Statement:** All relevant data are within the manuscript and its Supporting Information files(data).

**Funding:** The author(s) received no specific funding for this work.

## Abstract

To ensure the continuous high-efficiency operation of fuel cell systems, it is essential to perform real-time estimation of the maximum efficiency point and maximum power point for multi-stack fuel cell systems. The region between these two power points is commonly referred to as the "high-efficiency operating region." Initially, a transformation of the general expression for hydrogen consumption in multi-stack fuel cell systems is conducted to obtain an algebraic expression for the efficiency curve of multi-stack fuel cells. Utilizing a polynomial differentiation approach, the parameter equation for the maximum system efficiency is computed. Subsequently, a reverse deduction is carried out using the maximum efficiency and its corresponding power of underperforming subsystems to enhance the maximum efficiency of multi-stack fuel cell systems.Furthermore, an equivalent hydrogen consumption minimization method is introduced for real-time optimization of hybrid energy systems. The state machine control method serves as an auxiliary strategy, imposing the high-efficiency operating region as a boundary constraint for the equivalent hydrogen consumption minimization strategy's results. This ensures that the multi-stack fuel cell system operates as much as possible within the high-efficiency operating region.Through simulation validation using MATLAB/Simulink, the proposed approach comprehensively leverages the advantages of the state machine and equivalent hydrogen consumption. This approach enables effective identification of the high-efficiency operating region of fuel cells, while concurrently enhancing the operational range efficiency of the system, reducing hydrogen consumption, and elevating system stability.

## 1 Introduction

With the advancement of high-power fuel cell systems, mechanical failures leading to gas leakage may occur due to their large size. Additionally, non-uniform distribution of gases within internal reactions may result in poor voltage consistency among individual cells within the stack, thereby increasing operational complexities of the system [1]. In this context, multi-stack fuel cell systems (MFCS), as a viable solution for high-power applications, have a good

**Competing interests:** The authors have declared that no competing interests exist.

**Abbreviations:** Greek letters, ; ζ, empirical parameter; ξ, empirical parameter; θ, Actual value of the parameter to be estimated; η, efficiency; σ, standard deviation; θ̂, estimate of parameter; Subscripts, ; FCS, fuel cell system; SC, super-capacitor; hi, High; lo, Low; dis, Discharge; chg, Charge; opt, optimal; ref, Reference; Abbreviation, ; MFCS, multi-stack fuel cell systems; SOC, state of charge; ECMS, Equivalent Consumption Minimization Strategy; PEMFC, Proton exchange membrane fuel cell; SMC, State machine control policy; MFCS, Multi-stack fuel cell system.

performance in terms of space flexibility, system efficiency, and system energy consumption [2]. In addition, the use of MFCS helps to delay the degradation of fuel cells and enhance the stability of the system; When the fuel cell subsystem fails, the faulty system can be disconnected to form a degraded system to increase the reliability of the MFC system.

In 2020, Daimler Trucks AG [3] launched the Mercedes-Benz GenH2 hydrogen fuel cell truck, which uses two 150 kW fuel cells as power source and is equipped with two 40 kg liquid hydrogen storage tanks, with a total endurance of 1,000 kilometers. Coupled with a 72 kWh battery pack, it provides power support for the fuel cell in some vehicle scenarios. French Arles [4] built the world's first low-floor passenger train driven by fuel cells, the train can reach a maximum speed of 140km per hour, a one-time run of 1000 km, the kinetic energy generated during braking and excess fuel cell energy are stored in lithium batteries. Each car is equipped with a fuel cell and a hydrogen tank, and is equipped with a responsive intelligent energy management system and energy storage technology. In 2019, East Japan Railway Corporation (JR) [5] developed a new hybrid train, the FV-E991 series, consisting of two 180 kW fuel cell stacks and two 25 kWh lithium batteries, with an operating range of 140 km and a top speed of 100 km/h [6].

In order to realize the optimal energy management scheme of dual-pile fuel cell system and energy storage device, it is necessary to consider the fuel cell operating state and the SOC influence of supercapacitor. There is an "efficient operation zone" [7] in the efficiency diagram of the fuel cell system. Maintaining the output power of the fuel cell in this zone can optimize the efficiency and performance of the system. However, with the accumulation of fuel cell operating time, the dynamic characteristics of the fuel cell will change, and the range of efficient operating areas will also change. Therefore, real-time tracking of efficient operation area is one of the core issues of hybrid energy management system. As an energy storage device, ultracapacitors can cooperate with fuel cells through the charge/discharge process to make the fuel cells run sustainably and efficiently. In order to avoid overcharge and overdischarge of supercapacitors, constraints such as SOC boundary conditions and charge and discharge current need to be added to supercapacitors.

The strategy of minimum equivalent hydrogen consumption can minimize the hydrogen consumption of the system while meeting the power demand of the hybrid energy system [8], but this method cannot maintain the fuel cell system in the high-efficiency operation zone. As a rule control method, the state machine strategy can maintain the output power of supercapacitor SOC and fuel cell in the best state by making rules, but it relies too much on the designer's engineering experience, and the hydrogen consumption of the system cannot reach the optimal state.

In order to solve the above problems, the work of this paper is divided into two stages:

1. The general expression of maximum efficiency is obtained by first order derivation of multi-pile fuel cell efficiency expression. The expression is reversely verified and the optimal efficiency of non-homogeneous multi-pile fuel cells is modified, and the maximum efficiency of multi-pile fuel cells is estimated. The degradation experiment of fuel cell is predicted by grey prediction theory, and the characteristic value of the degradation curve is extracted, and then the degradation equation of fuel cell is established. Through the above two steps, the efficient operation area of the fuel cell can be obtained;

2. The obtained maximum efficiency interval is applied to the equivalent hydrogen consumption minimum strategy and the state machine strategy as constraint conditions. In order to synthesize the advantages of the two strategies, the state machine optimization method of the minimum hydrogen consumption strategy is proposed. The equivalent hydrogen consumption minimum strategy gives priority to the energy management results aiming at the

minimum hydrogen consumption, and then the state machine modifies the results according to the maximum efficiency constraint, so that the multi-stack fuel cell system can run in the high efficiency operation zone.

## 2 The energy management strategy for fuel cell hybrid power systems

### 2.1 The strategy of minimizing equivalent hydrogen consumption

The Equivalent Consumption Minimization Strategy (ECMS) was proposed by Xu Liangfei and others in 2002 and applied in fuel cell/hybrid electric buses. ECMS aims to reduce the instantaneous hydrogen consumption of fuel cell/hybrid electric systems. This method offers advantages such as effective control and minimal algorithm resource consumption. When the energy storage unit provides power compensation, the energy consumption of the hybrid energy system increases. However, during the charging of the energy storage unit, a portion of hydrogen is converted into electrical energy, leading to an overall reduction in the hydrogen consumption of the system.The algorithm is simplified by introducing the $R_{int}$ model of the power battery into the equivalent hydrogen consumption model. Therefore, from the perspective of total hydrogen consumption, the total instantaneous hydrogen consumption of the fuel cell / supercapacitor hybrid system of $C$sys is composed of $C_{FC}$ of the fuel cell and $C_{SC}$ of the supercapacitor. The objective function is:

$$\text{arminCsys(t)} = \min(f_1(P_{FC}(t)) + \lambda_{sc} f_2(P_{sc}(t))) \tag{1}$$

$$\text{s.t.} \begin{cases} SOC_{min} \leq SOC \leq SOC_{max} \\ I_{sc,min} \leq I_{sc} \leq I_{sc,max} \\ P_{FC,min} \leq P_{FC} \leq P_{FC,max} \\ -\Delta P_{FC} \leq \dfrac{dP_{FC}}{dt} \leq \Delta P_{FC} \end{cases} \tag{2}$$

In the above equation, $P_{FC,max}$ is the maximum output power of the fuel cell, $\frac{dP_{FC}}{dt}$ is the power variation of the fuel cell, $I_{sc,max}$ and $I_{sc,min}$ are the maximum charge/discharge current limits for the supercapacitor. SOC represents the ratio of the remaining capacity to the rated capacity of the energy storage device at a certain discharge rate, commonly expressed as a percentage, with a range of 0 to 1. When SOC = 0, it means the supercapacitor is fully discharged, and when SOC = 1, it means the supercapacitor is fully charged. $SOC_{max}$ is the upper limit of the supercapacitor's SOC, and $SOC_{min}$ is the lower limit of the supercapacitor's SOC, which is set to avoid overcharging or overdischarging of the supercapacitor, with an upper and lower limit of [0.3, 0.7]. $\lambda_{SC}$ is the penalty factor that is used to adjust the equivalent hydrogen consumption of the battery to minimize the deviation from the battery's target SOC. The penalty factor $\lambda_{SC}$ can be expressed as:

$$\lambda_{SC} = 1 - \frac{2\mu(SOC(t) - 0.5(SOC_{max} + SOC_{min}))}{SOC_{max} + SOC_{min}} \tag{3}$$

$\lambda_{SC}$ is limited by the coefficient µ to keep the SOC of the supercapacitor within the range [$SOC_{min}$, $SOC_{max}$] under certain operating conditions. The equivalent hydrogen consumption

of the supercapacitor $C_{SC}$ is calculated as follows:

$$C_{SC} = \begin{cases} \dfrac{P_{SC}}{\eta_{dis}(t)\bar{\eta}_{chg}\eta_{dc/dc}}\dfrac{C_{fcs,avg}}{P_{fcs,avg}}, & P_{sc} \geq 0 \\[4mm] P_{SC}\eta_{chg}(t)\bar{\eta}_{dis}\eta_{dc/dc}\dfrac{C_{fcs,avg}}{P_{fcs,avg}}, & P_{sc} < 0 \end{cases} \tag{4}$$

In the above equation, $P_{SC}$ represents the output power of the supercapacitor, with positive and negative values indicating the charging and discharging states of the supercapacitor; $C_{fcs,avg}$ represents the average hydrogen consumption of the fuel cell; $P_{fcs,avg}$ represents the average power of the fuel cell system; $\frac{C_{fcs,avg}}{P_{fcs,avg}}$ represents the reciprocal of the fuel cell system's average efficiency $\frac{1}{\eta_{fcs}}$, which is considered a constant in the actual calculation process; $\bar{\eta}_{chg}$ and $\bar{\eta}_{dis}$ are the average charging and discharging efficiency of the supercapacitor, respectively. In this paper, a first-order RC model is used as the efficiency model for the supercapacitor system, which can be analogized with the $R_{int}$ model of the battery. In the first-order RC model, the resistance value of the supercapacitor is small and can be regarded as a constant, therefore, the charging and discharging efficiency of the supercapacitor can be simplified as follows:

$$\eta_{SC}(t) = \begin{cases} \eta_{dis}(t) = \dfrac{\left(1 + \sqrt{1 - \dfrac{4R_{SC}P_{SC}(t)}{U_{OCV}^2}}\right)}{2} \\[6mm] \eta_{chg}(t) = \dfrac{2}{\left(1 + \sqrt{1 - \dfrac{4R_{SC}P_{SC}(t)}{U_{OCV}^2}}\right)} \end{cases} \tag{5}$$

In the above equation, $P_{SC}$ is the charging and discharging power of the supercapacitor, $R_{SC}$ is the internal resistance of the supercapacitor, and $U_{ocv}$ is the terminal voltage of the supercapacitor. The relationship between the power of the supercapacitor and the power of the fuel cell is as follows:

$$P_{load} = P_{SC} + P_{MFC} \tag{6}$$

According to Eq (4), the simplified expression for the equivalent hydrogen consumption of the supercapacitor is as follows:

$$C_{SC} = \frac{P_{SC} \cdot \sigma}{LHV \cdot \eta_{FC\_avg}} \tag{7}$$

In the equation, LHV represents the lower heating value of hydrogen (120 MJ/kg), $\eta_{FC\_avg}$ represents the average efficiency of the fuel cell, where $\sigma$ is defined a s follows:

$$\sigma = \begin{cases} \dfrac{1}{\eta_{dis}(t)\bar{\eta}_{chg}} & Psc \geq 0 \\[4mm] \eta_{chg}(t)\bar{\eta}_{dis} & Psc < 0 \end{cases} \tag{8}$$

Based on Eqs (4) and (7), further simplification is made for $C_{SC}$ by defining the variable K.

$$K = \frac{\lambda_{SC} \cdot \sigma}{LHV \cdot \eta_{FC\_avg}} \tag{9}$$

The energy consumption function of each fuel cell stack in a multiple stack fuel cell system

is given by:

$$C_{FCsn} = a_n P_{FCSn}^2 + b_n P_{FCSn} + c_n \tag{10}$$

In the equation, $C_{FCsn}$ represents the hydrogen consumption of the stack, $P_{FCSn}$ represents the real-time power of the stack, and $a_n$、$b_n$、$c_n$ are the second-order fitting parameters for the hydrogen consumption and power of the stack.

By using Eqs (1), (7) and (9), the relationship between the total instantaneous hydrogen consumption of the hybrid power system $C_{MFC,SC}$ and the net output power of the fuel cell $P_{MFC}$ can be obtained.

$$C_{MFC,SC} = \frac{a_1 a_2}{a_1 + a_2} P_{MFC}^2 + \left( \frac{b_1 a_2 + b_2 a_1}{a_1 + a_2} - \frac{K\sigma}{LHV\eta_{FC,avg}} \right) P_{MFC} + (c_1 + c_2) - \frac{(b_1 - b_2)^2}{4(a_1 + a_2)}$$
$$+ \frac{K\sigma P_{load}}{LHV\eta_{FC,avg}}) \tag{11}$$

Through Eq (11), the second-order partial derivative of $C_{MFC,SC}$ with respect to $P_{MFC}$ is satisfied.

$$\frac{d^2 C_{sys}}{dP_{MFC}^2} = \frac{2a_1 a_2}{a_1 + a_2} > 0 \tag{12}$$

Therefore, Eq (10) has a minimum value, which is the optimal output power of the fuel cell system $P_{MFC,opt}$. The expression is:

$$P_{MFC,opt} = \frac{a_1 + a_2}{2a_1 a_2} \cdot \left( \frac{K\sigma}{LHV\eta_{FC,avg}} - \frac{b_1 a_2 + b_2 a_1}{a_1 + a_2} \right) \tag{13}$$

In summary, the optimal power allocation strategy for fuel cells and supercapacitors is as follows:

$$\begin{cases} P_{MFC} = P_{MFC,opt} \\ P_{SC} = P_{load} - P_{MFC,opt} \end{cases} \tag{14}$$

## 2.2 Optimization of power allocation strategy for fuel cell hybrid energy system in high-efficiency area

The efficiency chart of the fuel cell system is shown in Fig 1. It demonstrates the variation of efficiency with load power and is one of the key characteristics of the fuel cell system. The system efficiency gradually increases with an increase in load power after system startup, reaching a maximum value. It then slightly decreases until the system reaches its maximum output power. The system operation is typically divided into two stages: the "inefficient region" and the "high-efficiency operation region" (also known as the "Safe Operating Zone"). The "inefficient region" refers to the initial stage after the fuel cell system starts, where the "parasitic power consumption" of auxiliary devices accounts for a significant proportion of the stack power, resulting in lower overall system efficiency. Therefore, this region should be avoided as much as possible when designing energy management strategies. As the load power increases, the fuel cell output power exceeds the parasitic power consumption, leading to a decrease in its proportion in the stack power. This stage is referred to as the "high-efficiency operation region." In the efficiency chart, the point corresponding to the maximum efficiency is denoted

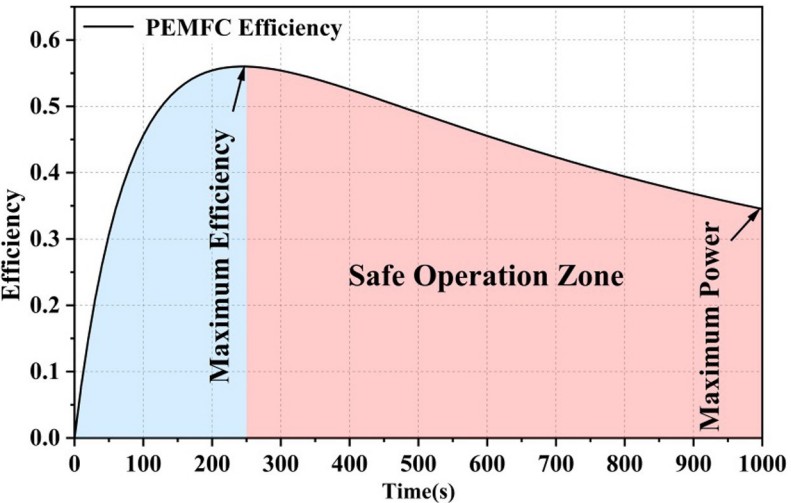

**Fig 1. Map of PEMFC system efficiency zones.**

as $\eta_{max}$, and the load power at this point is referred to as the maximum efficiency operating point, denoted as $P_{eff,max}$. The maximum power point of the system is denoted as $P_{fcs,max}$. The two operating regions are divided by the maximum efficiency point ($\eta_{max}$). The "inefficient region" ranges from load power "0" to the maximum efficiency operating point $P_{eff,max}$, inclusive ($[0,P_{eff,max}]$). The "high-efficiency operation region" is determined by $P_{eff,max}$ and $P_{fcs,max}$ and ranges from $P_{eff,max}$ to $P_{fcs,max}$, inclusive $[P_{eff,max},P_{fcs,max}]$. As the fuel cell degrades during operation, $P_{fcs,max}$ and $P_{eff,max}$ also change, requiring real-time updates during system operation.

Feroldi [9] proposed using a state machine policy to ensure that the fuel cell in the hybrid energy system operates in the high-efficiency region, as shown in Fig 2. In consideration of the fuel cell efficiency range and the SOC (state of charge) status of the supercapacitor, the hydrogen consumption is minimized to ensure that the fuel cell system operates in the optimal efficiency range. $P_{fcs,lo}$ and $P_{fcs,max}$ represent the lower and upper bounds of the power in the

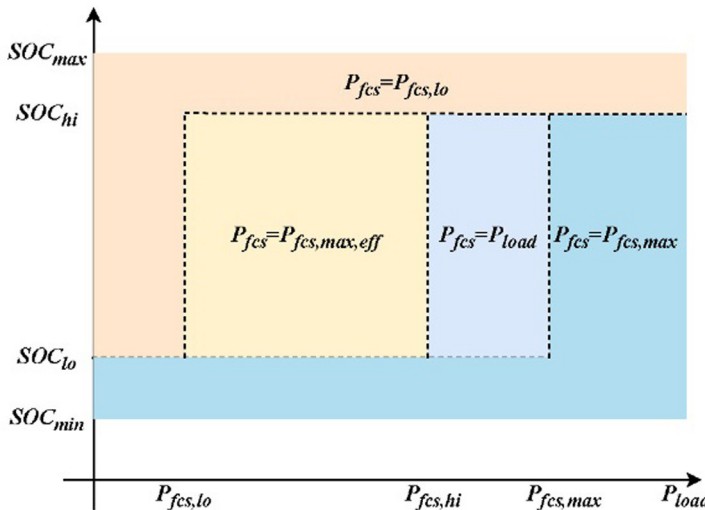

**Fig 2. Power allocation strategy for fuel cell hybrid systems based on efficiency maps.**

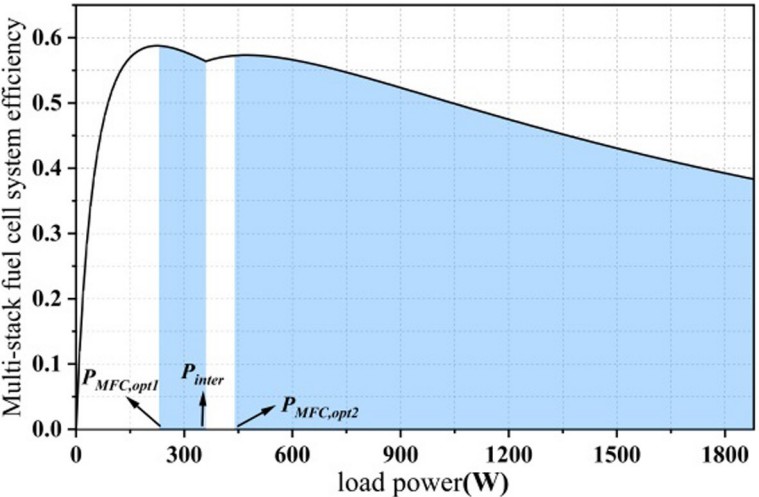

**Fig 3. Distribution of "high efficiency zones" for dual stack fuel cells.**

"high-efficiency operation region" of the fuel cell, respectively. As for the energy storage device, its SOC is divided into four regions, including $SOC_{min}$, $SOC_{max}$, $SOC_{hi}$, and $SOC_{lo}$, $SOC_{min}$ and $SOC_{max}$ denote the upper and lower bounds of the energy storage device, while $SOC_{hi}$ and $SOC_{lo}$ represent higher and lower states of charge, respectively.[$SOC_{min}$, $SOC_{lo}$] and [$SOC_{hi}$, $SOC_{max}$] are used as critical buffer regions in the energy storage device to avoid frequent power charging and discharging caused by a single critical value. In [$SOC_{min}$, $SOC_{lo}$] the energy storage device has a lower state of charge and should not output power, while the fuel cell should charge it. In [$SOC_{hi}$, $SOC_{max}$], the energy storage device can output power together with the fuel cell due to its high SOC. In [$SOC_{lo}$, $SOC_{hi}$], the energy storage device can reduce its workload appropriately.

Focus on investigating the power allocation strategy in partial startup mod, as this mode transitions to full startup mode when the load power exceeds the inflection point. Conducting research on the partial startup mode allows for a comprehensive calculation of efficiency optima in multi-stack fuel cell systems [10].

The efficiency chart of multi-stack fuel cell systems exhibits multiple peaks due to the presence of the inflection point. Consequently, the 'safe operating region' for a dual-stack fuel cell will also be divided into two, as depicted in Fig 3. Thus, within this efficiency range, it is essential to consider the two highest efficiency points, their corresponding power levels, the inflection point, and the maximum power point as parameters for the state-machine control strategy.

In a hybrid energy system, to ensure that the fuel cell operates in the "high-efficiency region", it is also necessary to maintain the SOC of the supercapacitor. In consideration of these factors, the improved state machine policy is shown in Fig 4.

$P_{fc,opt1}$ represents the system power corresponding to the highest efficiency point of the partial start-up mode in a multi-stack fuel cell system, and $P_{fc,opt2}$ represents the system power corresponding to the maximum efficiency point of the full start-up mode. $P_{fc,inter}$ represents the intersection point of the partial start-up mode and the full start-up mode, at which both modes have the same efficiency and hydrogen consumption rate. [0, $P_{fc,opt1}$] represents the start-up stage of the fuel cell, during which the power mainly supplies the auxiliary consumption (control, fans, etc.). The efficiency in this region is relatively low, so it should be avoided during system operation [11].

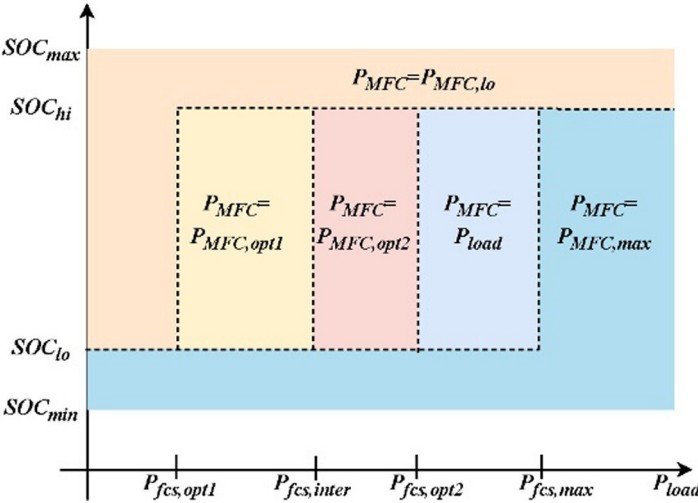

**Fig 4. MFC improvement strategies based on efficiency diagrams.**

The modified state machine policy is as follows:

When $SOC \epsilon [SOC_{hi}, SOC_{max}$, where the supercapacitor SOC is relatively high and can be used as the primary power source for output. Therefore, the power allocation strategy is as follows:

$$P_{MFC,ref} = \begin{cases} P_{MFC,lo} & if\ P_{MFC} < P_{MFC,lo} \\ P_{MFC,lo} & if\ P_{MFC,lo} < P_{MFC} < P_{MFC,max} \\ P_{MFC,Max} & if\ P_{MFC} > P_{MFC,max} \end{cases} \quad (15)$$

When $SOC \epsilon [SOC_{lo}, SOC_{hi}]$, the power output of the supercapacitor and fuel cell should be determined based on the load demand.

$$P_{MFC,ref} = \begin{cases} P_{MFC,op1} & if\ P_{MFC} < P_{MFC,opt1} \\ P_{load} & if\ P_{MFC,opt1} \le P_{MFC} \le P_{MFC,inter} \\ P_{MFC,opt2} & if\ P_{MFC,inter} < P_{MFC} < P_{MFC,opt2} \\ P_{load} & if\ P_{MFC,opt2} \le P_{MFC} < P_{MFC,max} \\ P_{MFC,max} & if\ P_{MFC} \ge P_{MFC,max} \end{cases} \quad (16)$$

When the $SOC \in [SOC_{min}, SOC_{lo}]$, the supercapacitor SOC is low. The supercapacitor does not work, and the fuel cell serves as the main power source. To meet the load power demand, it is necessary to charge the supercapacitor, taking into account the maximum charging current of the supercapacitor and the maximum power fluctuation $\Delta P_{MFC}$ of the fuel cell. The efficiency map is revised again, and the state machine partition results are as follows.

$$P_{MFC,ref} = \begin{cases} P_{MFC,op1} & if\ P_{MFC} < P_{MFC,opt1} \\ P_{MFC,inter} & if\ P_{MFC,opt1} \le P_{MFC} \le P_{MFC,inter} \\ P_{MFC,opt2} & if\ P_{MFC,inter} < P_{MFC} < P_{MFC,opt2} \\ P_{load} + \Delta P_{MFC} & if\ P_{MFC,opt2} \le P_{MFC} < P_{MFC,max} \\ P_{MFC,max} & if\ P_{MFC} \ge P_{MFC,max} \end{cases} \quad (17)$$

## 2.3 Power allocation strategy based on equivalent minimum hydrogen consumption-state machine

The core steps of ECMS involve converting the electrical energy consumption of energy storage elements into equivalent hydrogen consumption [12]. The primary aim is to reduce the hydrogen consumption of hybrid power systems while providing reference values for the optimal charge and discharge power of supercapacitors. However, the equivalent minimal hydrogen consumption strategy introduces constraints to accommodate fuel cell transient changes. Consequently, when the load power is high, ECMS calculations may conflict with constraint conditions, leading to the ineffectiveness of ECMS.

The purpose of the state machine is to determine the reference power of the fuel cell when there is a change in state. It possesses advantages such as high reliability, ease of implementation, and robustness. However, in practical system operation, due to the inability to adjust predefined rules online, the dynamic characteristics of the fuel cell change as the system operates, resulting in a deterioration of control strategy performance. Additionally, insufficient designer experience may lead to control logic errors, causing equipment to malfunction.The ECMS is combined with the state machine method to form a hierarchical energy management strategy. The upper layer strategy solves the energy management problem between the multi-reactor fuel cell system and the ultracapacitors based on the state machine strategy with equivalent minimum hydrogen consumption. In this layer policy, Multi-stack fuel cell system is seen as a high-power fuel cell system, The calculation result is the optimized power $P_{dual,sys}$ of the two-reactor fuel cell system, $P_{dual,sys}$ is entered into the lower-level policy, The lower layer strategy is the power allocation strategy for the multi-reactor fuel cell system, Multi-stack fuel cell power distribution strategy for $P_{dual,sys}$ for the power distribution, By collecting parameters such as the fuel cell system power and hydrogen consumption, Update the parameters of the power consumption model in the power allocation strategy after calculation, The strategy architecture diagram is shown in Fig 5, The updated system efficiency maximum value and the corresponding power value are transmitted into the upper layer policy for parameter update.

The reference power $P_{MFC,ref}$ of a multi-stack fuel cell system depends on the power demand, the SOC of the supercapacitor, and $P_{MFC,opt}$ and is defined according to the SOC state as follows:

(1) State 1: When the supercapacitor $SOC > SOC_{hi}$, the supercapacitor has a higher discharge power, making the fuel cell system more concentrated in the high-efficiency operating zone.

$$P_{MFC,ref} = \begin{cases} P_{fc,min} & if\ P_{load} < P_{fc,min} \\ P_{load} & if\ P_{fc,min} < P_{load} < P_{fc,max} \\ P_{fc,max} & if\ P_{load} \geq P_{fc,max} \end{cases} \tag{18}$$

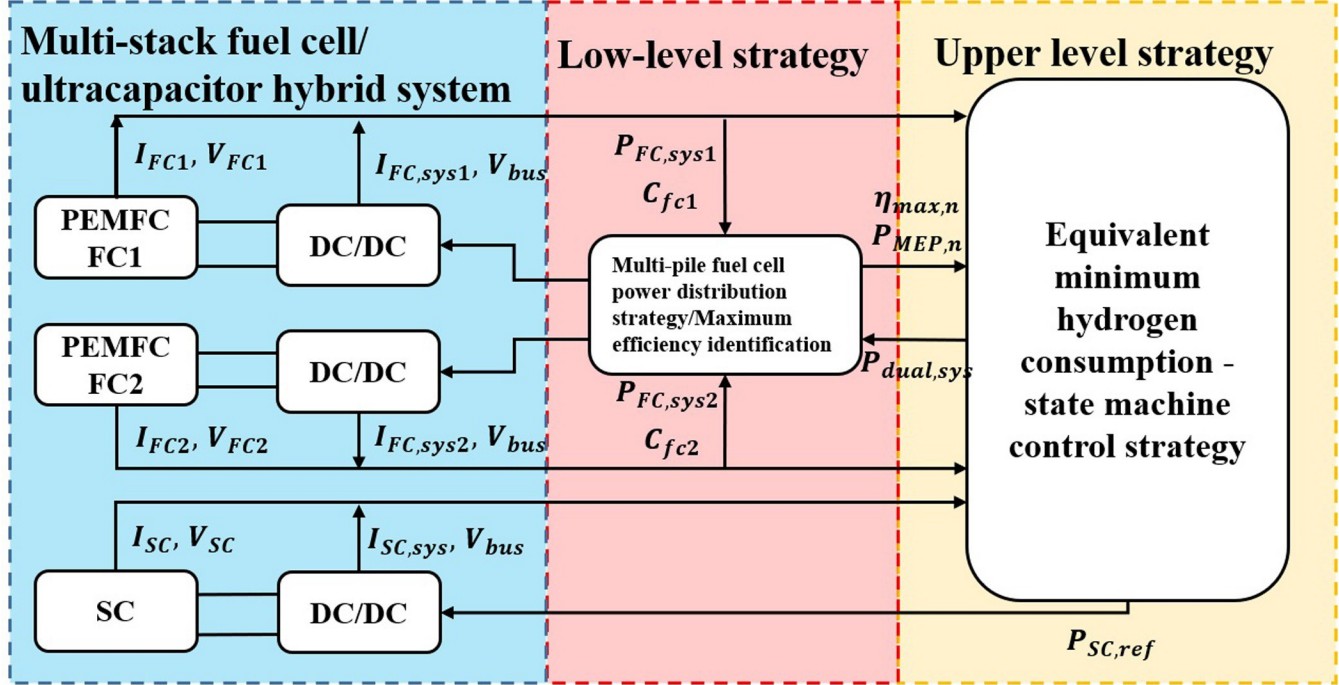

**Fig 5. Architecture diagram of a hybrid fuel cell/supercapacitor system based on minimisation of equivalent consumption.**

(2) State 2: When the SOC of the supercapacitor is in the range of [SOC$_{lo}$,SOC$_{hi}$].

$$P_{MFC,ref} = \begin{cases} P_{fc,min} & if\ P_{load} < P_{fc,min} \\ P_{fc,opt} & if\ P_{fc,min} < P_{load} < P_{fc,opt} \\ P_{load} & if\ P_{fc,opt} \leq P_{load} < P_{fc,max} \\ P_{fc,max} & if\ P_{load} \geq P_{fc,max} \end{cases} \tag{19}$$

(3) State 3: When the SOC of the supercapacitor is in the range of [SOC$_{min}$,SOC$_{lo}$].

$$P_{MFC,ref} = \begin{cases} P_{load} + P_{SC,char} & if\ P_{load} < P_{fc,min} \\ \max\left(P_{load} + \Delta P_{MFC}, P_{fc,opt}\right) & if\ P_{fc,min} \leq P_{load} < P_{fc,opt} \\ P_{fc} + \Delta P_{MFC} & if\ P_{fc,opt} \leq P_{load} < P_{fc,max} \\ P_{fc} = P_{fc} + \Delta P_{MFC} & if\ P_{load} \geq P_{fc,min} \end{cases} \tag{20}$$

Eqs (1)–(20) form an equivalent minimum hydrogen consumption-state machine power allocation strategy, as shown in Fig 6. The ECM calculates the instantaneous minimum hydrogen consumption power P$_{MFC,opt}$ of the MFC based on the load power and the SOC of the supercapacitor. The SMC modifies P$_{MFC,opt}$ based on the SOC state of the supercapacitor, the maximum efficiency point P$_{MFC,MEPn}$ and the maximum power point P$_{MFC,max}$ as reference values, and calculates the power reference values P$_{MFC,ref}$ and P$_{SC,ref}$ for the multi-stack fuel

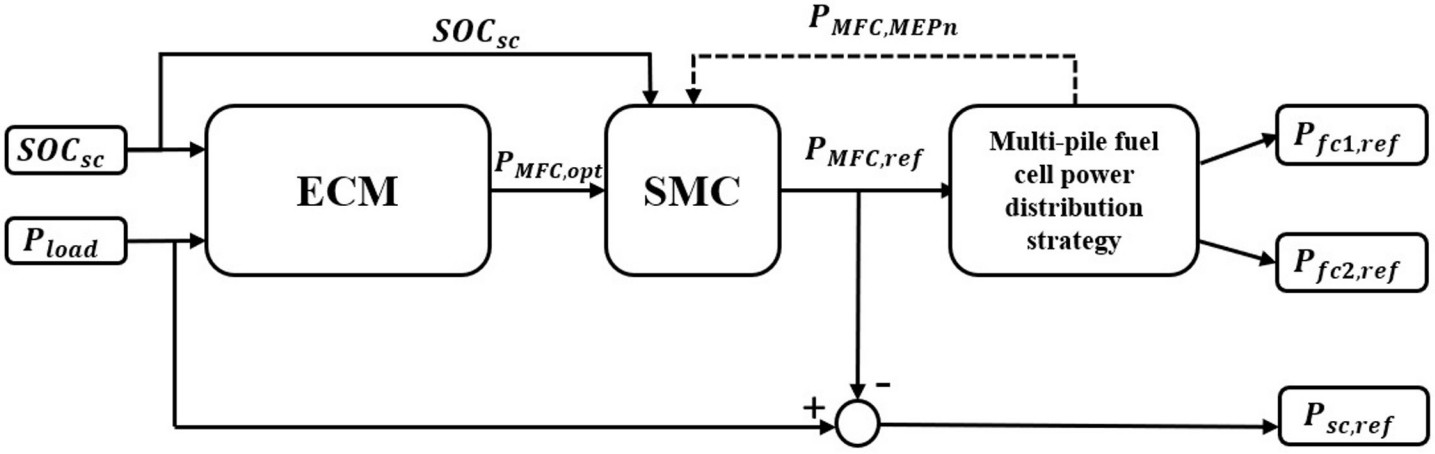

**Fig 6. Equivalent minimum hydrogen consumption - state machine power allocation strategy.**

cell system and the supercapacitor. $P_{MFC,ref}$ is used to distribute power to the fuel cell subsystem through a multi-stack fuel cell power allocation strategy.

## 3 Maximum efficiency estimation for dual-stack fuel cell systems

According to Eqs (18) to (20), in order to maintain the output power of the fuel cell system within the high-efficiency operating range, real-time tracking of the fuel cell system's maximum efficiency point is necessary [13].

### 3.1 Calculation of efficiency extremes for dual-stack fuel cell systems

In partial load modes of power allocation strategies for multi-stack fuel cell systems, when the load power doesn't exceed the inflection power, the system will activate fewer subsystems to respond to the load power [14]. Therefore, during the calculation of efficiency extremes, efficiency calculations are still based on the inflection power.

The inflection power for dual-stack fuel cells is denoted as:

$$P_{inter} = \frac{2\sqrt{c_2(a_1 + a_2)} - (b_1 - b_2)}{2a_1} \tag{21}$$

When $P_{load} > P_{inter}$, both fuel cell subsystems will start up, and the power allocation strategy and hydrogen consumption of the dual-stack system are shown in Eq (22).

$$\begin{cases} P_{FC1} = \dfrac{a_2 P_{MFC}}{a_1 + a_2} - \dfrac{b_1 - b_2}{2(a_1 + a_2)}, P_{FC2} = \dfrac{a_1 P_{MFC}}{a_1 + a_2} + \dfrac{b_1 - b_2}{2(a_1 + a_2)} \\ C_{MFC} = \dfrac{a_1 a_2}{a_1 + a_2} P_{MFC}^2 + \dfrac{b_1 a_2 + b_2 a_1}{a_1 + a_2} P_{MFC} + c_1 + c_2 - \dfrac{(b_1 - b_2)^2}{4(a_1 + a_2)} \end{cases} \tag{22}$$

Based on the fuel cell energy consumption function (10) and the general expression for hydrogen consumption in the dual-stack fuel cell system (22), the efficiency expression of the

dual-stack fuel cell system is as follows.

$$\begin{cases} \eta_{MFC} = \dfrac{P_{MFC}}{aP_{MFC}^2 + bP_{MFC} + c} & P_{MFC} \le P_{inter} \\[4mm] \eta_{MFC} = \dfrac{P_{MFC}}{\dfrac{a_1 a_2}{a_1 + a_2} P_{MFCS}^2 + \dfrac{b_1 a_2 + b_2 a_1}{a_1 + a_2} P_{MFCS} + c_1 + c_2 - \dfrac{(b_1 - b_2)^2}{4(a_1 + a_2)}} & P_{MFC} > P_{inter} \end{cases} \quad (23)$$

Using Eqs (21) and (23), the efficiency of the system at the crossover power $P_{inter}$ is 56.26%. According to the system efficiency characteristics of the multi-stack fuel cell, there are two other load power points in the efficiency graph that have the same efficiency as the crossover power, denoted as $P_1$ and $P_2$.

Fig 7 represents the distribution of $P_1, P_2$, and their corresponding efficiency points in the efficiency graph of the dual-stack fuel cell system. In both the partial startup mode and the full startup mode of the multi-stack fuel cell, it is observed that $\eta_{MFC}(P_1) = \eta_{MFC}(P_{inter})$ and $\eta_{MFC}(P_{inter}) = \eta_{MFC}(P_1)$. According to the Rolle's theorem, there must exist $\xi_1 \in (P_1, P_{inter})\xi_2 \in (P_{inter}, P_1))$ and $\xi_2 \in (P_{inter}, P_1)$ within this range, such that $\eta'_{MFC}(\xi_1) = \eta'_{MFC}(\xi_2) = 0$. Therefore, the maximum efficiency point is derivable.

$$\frac{d\eta_{MFC}}{dP_{MFC}} = 0 \quad (24)$$

Solving Eqs (23) and (24), it obtains the maximum efficiency result of the dual-stack fuel cell system:

$$\begin{cases} P_{MFC} = \sqrt{\dfrac{c}{a}} & P_{MFC} \le P_{inter} \\[4mm] P_{MFC} = \sqrt{\dfrac{(c_1 + c_2)(a_1 + a_2) - (b_1 + b_2)^2}{a_1 a_2}} & P_{MFC} > P_{inter} \end{cases} \quad (25)$$

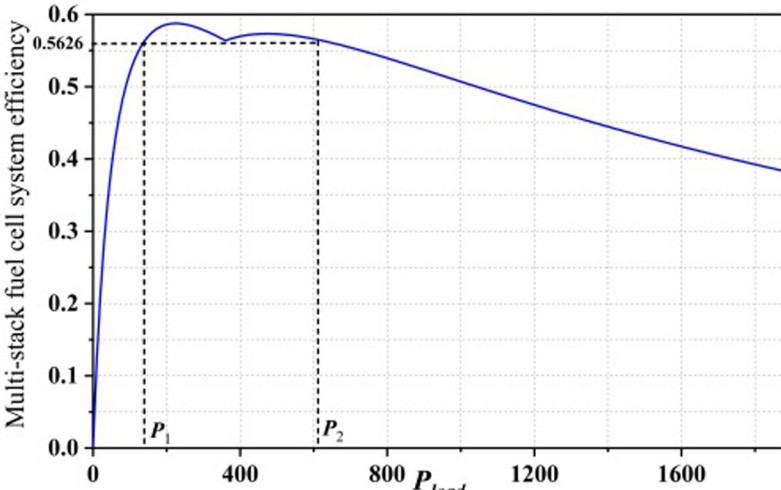

**Fig 7. Fuel cell system efficiency versus power curve.**

### 3.2 The estimation result of maximum efficiency

Solving Eq (25) gives the maximum efficiency and the estimated maximum efficiency values of the PEMFC1 and PEMFC2 subsystems of the dual-stack fuel cell system, as shown in Fig 8.

The maximum efficiency point and the corresponding power point of the partial startup mode of the dual-stack fuel cell system are $\left[P_{\mathrm{MEP1}}, \eta_{\mathrm{MEP1}}\right] = (224.74, 58.10\%)$ The maximum efficiency point and the corresponding power point of the full startup mode are $\left[P_{\mathrm{MEP2}}, \eta_{\mathrm{MEP2}}\right] = (437.93, 56.01\%)$. After power allocation, the efficiency points and corresponding power loads of PEMFC1 and PEMFC2 at the maximum efficiency point in the full startup mode are $\left[P_{\mathrm{FC1,ME}}, \eta_{\mathrm{FC1,ME}}\right] = (219.12, 58.75\%)$ and $\left[P_{\mathrm{FC2,ME}}, \eta_{\mathrm{FC2,ME}}\right] = (218.8, 55.83\%)$.

In the partial startup mode, the dual-stack fuel cell system activates only one fuel cell stack, which results in the system's maximum efficiency point being the same as that of a single-stack fuel cell. However, in the full startup mode, when the dual-stack fuel cell system operates within the high-efficiency region, it is essential to calculate and compare the maximum efficiency points of both the multi-stack fuel cell system and its subsystems. This comparison helps verify whether the subsystems also operate within the high-efficiency region when the entire system is in high-efficiency operation.

When the dual-stack fuel cell system operates at the maximum efficiency point in the full startup mode, the comparison results of power points and corresponding maximum efficiency points of the subsystems in the efficiency map are shown in Figs 9 and 10. The red dashed line represents the subsystem power and efficiency corresponding to the maximum efficiency point $P_{\mathrm{MFC,opt2}}$ in the full startup mode, while the black dashed line represents the maximum efficiency point and corresponding power of a single fuel cell. From the figure, it can be seen that when the dual-stack fuel cell system operates at the maximum efficiency point, the subsystems are not operating at their optimal states.

In order to ensure that the operating states of all subsystems in the multi-stack fuel cell system can be maintained within the "high-efficiency operating region", the corresponding powers $P_{\mathrm{MFC,opt2,1}}$ and $P_{\mathrm{MFC,opt2,2}}$ of the PEMFC1 and PEMFC2 subsystems at their respective maximum efficiency operating points are derived in reverse using [15] Eq (26). The calculation

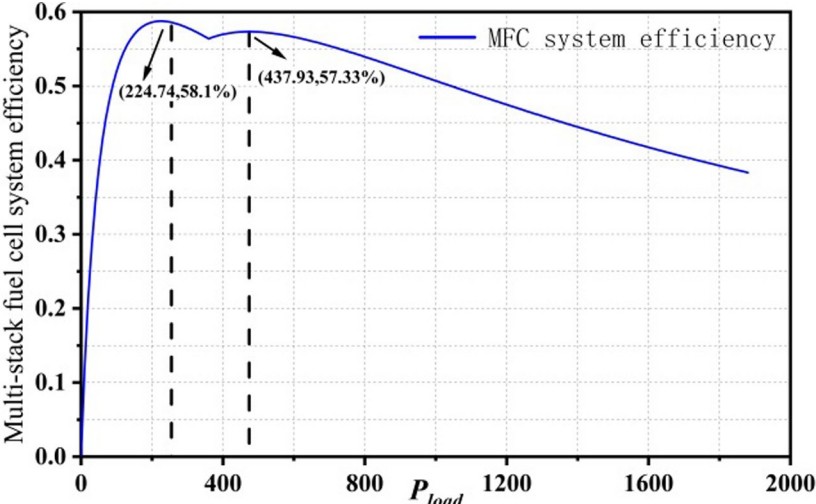

**Fig 8. Max Efficiency points for dual stack fuel cell systems.**

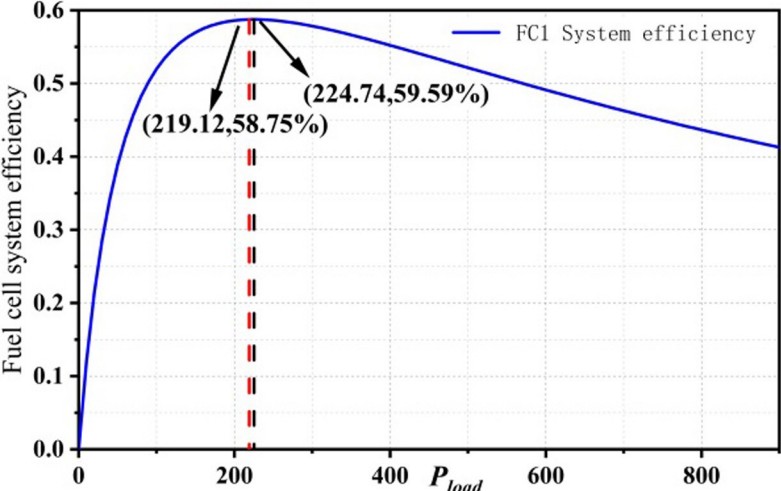

**Fig 9. ME comparison in FC1.**

process is as follows:

$$P_{MFC,opt2,1} = \left( P_{FC1,MEP} - \frac{b_2 - b_1}{2(a_1 + a_2)} \right) * \frac{(a_1 + a_2)}{a_2}$$

$$P_{MFC,opt2,2} = \left( P_{FC2,MEP} + \frac{b_2 - b_1}{2(a_1 + a_2)} \right) * \frac{(a_1 + a_2)}{a_1}$$

(26)

The calculation of $\eta_{MEP2}$, based on Eqs (24) and (26) confirms the estimation of the maximum efficiency points for the MFC system, PEMFC1, and PEMFC2 using $P_{MEP2}$, $P_{MEP2}$, and $P_{MFC,opt2,2}$. The results indicate that even after applying power allocation strategies, the fuel cell subsystems do not operate entirely within the high-efficiency operating region. This is because the multi-stack allocation strategy used in the fuel cell paper does not result in uniform distribution, and the significant degradation differences between fuel cells cause a deviation

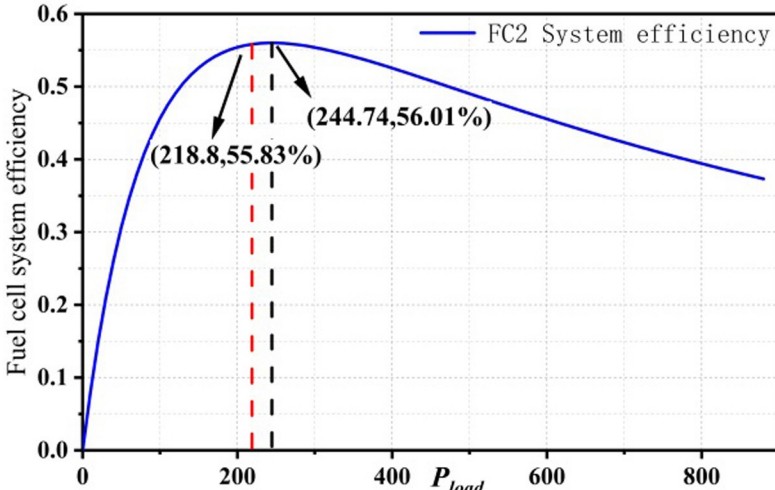

**Fig 10. ME comparison in FC2.**

**Table 1. Comparison of results of maximum efficiency points.**

|   | $P_{MEP2}$ | $\eta_{MEP2}$ | $P_{FC1,MEP}$ | $\eta_{FC1,MEP}$ | $P_{FC2,MEP}$ | $\eta_{FC2,MEP}$ |
|---|---|---|---|---|---|---|
| 1 | 437.93 | 57.33% | 219.12 | 58.75% | 218.80 | 55.83% |
| 2 | 448.39 | 57.29% | 224.74 | 59.59% | 222.90 | 55.89% |
| 3 | 499.34 | 57.30% | 254.60 | 58.59% | 244.74 | 56.01% |

from the actual optimal efficiency point. The efficiency comparison results of the dual-stack fuel cell system and subsystems can be calculated using $P_{MFC,ME2}$, $P_{FC2,MEP}$ and $P_{FC2,MEP}$ as follows:

The data in groups 1–3 of Table 1 represent the net output power and system efficiency of the multi-stack fuel cell and its subsystems when the MFC, PEMFC1, and PEMFC2 operate at their respective maximum efficiency power points. The results indicate that when the fuel cell subsystem FC2 operates at its maximum efficiency point, both the multi-stack fuel cell system and its subsystems operate within the high-efficiency range. Given the performance differences between PEMFC2 and PEMFC1 in practice, the maximum efficiency of the dual-stack fuel cell should be determined by the lower-performing stack.Through reverse engineering via the $P_{FC2,MEP}$ method, the power corresponding to the maximum efficiency of the MFC system is determined to be (499.34, 57.30%). Although the directly derived system efficiency for the multi-stack fuel cell is slightly lower, this method ensures that all subsystems can operate at their optimal power levels, contributing to the overall efficient power output of the fuel cell.

## 4 Analysis of simulation results

Fig 11 represents the test load curve, with a continuous load duration of approximately 400 seconds and a load power range of 0-1100W. It consists of three repetitive load cycles, used to test the three power allocation strategies mentioned in this paper. Table 2 outlines the various constraint conditions within the energy management strategy.

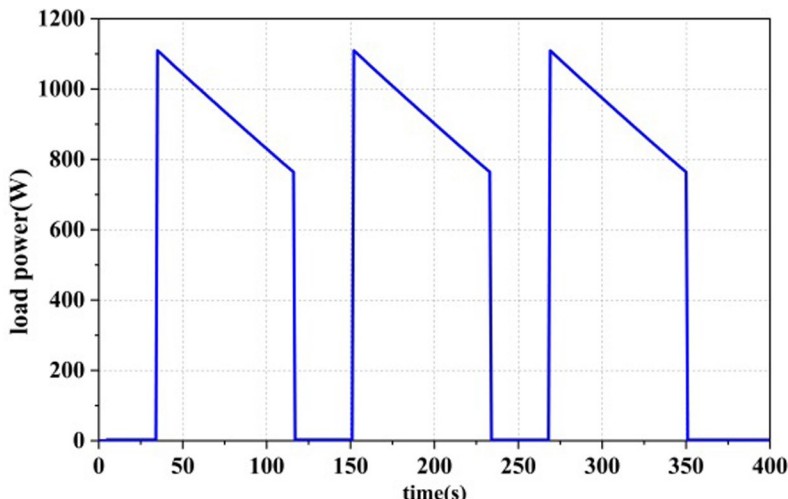

**Fig 11. Power load curves.**

Table 2. Constraints in hybrid energy systems.

| Parameter | Value |
|---|---|
| $\mu$ | 0.6 |
| $P_{MFC,min}$(W) | 0 |
| $P_{MFC,lo}$(W) | 100 |
| $P_{MFC,max}$(W) | 1880 |
| $\|\Delta P_{FC}\|$(W/s) | 300 |
| $SOC_{min}$(%) | 20 |
| $SOC_{lo}$(%) | 30 |
| $SOC_{hi}$(%) | 70 |
| $SOC_{max}$(%) | 80 |

## 4.1 Power allocation results

As shown in Figs 12–14, all three power allocation strategies are capable of meeting the power requirements of the load. During system operation, the dual-stack fuel cell can provide stable power to the system. The supercapacitor, serving as an auxiliary energy source, can accommodate the frequent fluctuations in load demand, compensate for power shortages during operation, and recover excess power.

Three power allocation strategies were implemented with a minimum output power set for the fuel cell. Even when there was no power load at the beginning of system operation, the fuel cell still operated at the minimum power. Therefore, in the first 25 seconds, the fuel cell output power was 100W, and the supercapacitor power was negative. In the SMC strategy, the supercapacitor served as the primary output with a high SOC exceeding $SOC_{hi}$, as shown in Fig 13, and as the supercapacitor SOC decreased, the fuel cell would take on the load power as the main power source. In the ECM and ECM+SMC strategies, the first half of the strategies were consistent. As the supercapacitor SOC decreased, in order to ensure the SOC of the supercapacitor, when there was no load output, the fuel cell would operate at the maximum efficiency corresponding to the power P_opt1 by correcting its state with the state machine strategy, as shown in Figs 12 and 14.

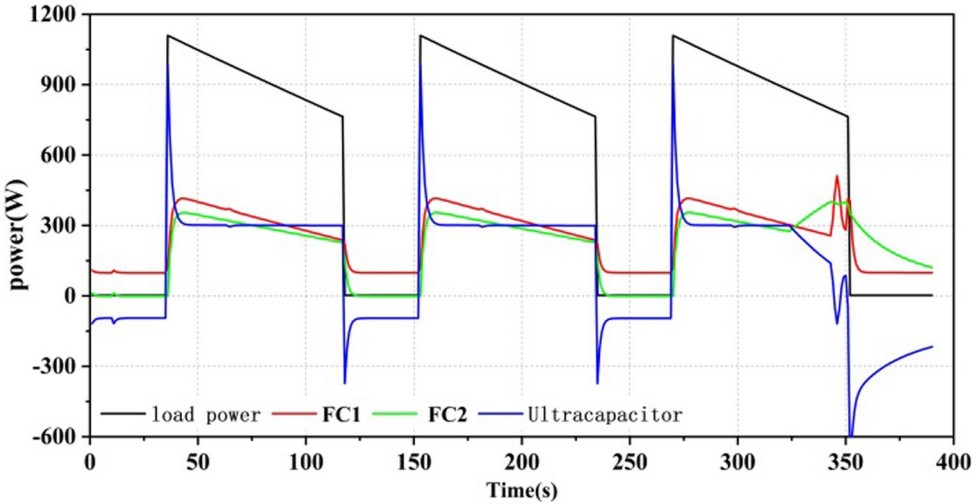

Fig 12. Equivalent hydrogen consumption energy allocation results.

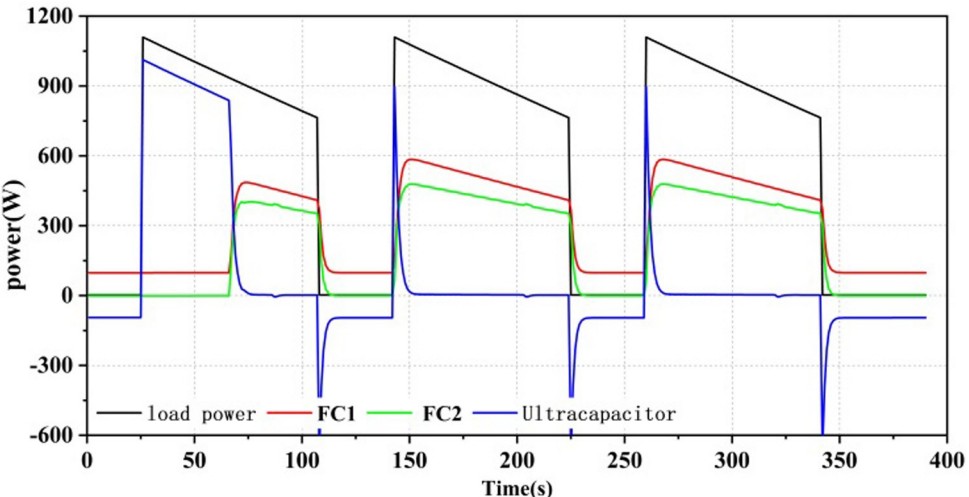

**Fig 13. State machine control energy allocation results.**

## 4.2 Efficiency comparison of dual-stack systems

Under different control strategies, the efficiency curve of the dual-stack fuel cell system is shown in Figs 15–17, where the system efficiency $\eta_{dual,sys}$ is calculated according to Eq (3). The green area represents the high-efficiency range in which the dual-stack fuel cell system operates, in order to display the optimization results [16]. Under all three strategies, the system efficiency drops abruptly to different degrees due to the sudden increase in fuel cell system power, As evident from the load curve in Fig 11, peak values in the efficiency curve occur simultaneously with moments of sudden load increase or decrease. The generation of these peaks is closely related to the output power of the fuel cell. Particularly within the low-power range, the system efficiency increases with increasing power, hence the occurrence of multiple peaks in the efficiency curve. Among the three power allocation strategies, the ECM-SMC strategy has the largest area under the high-efficiency operating region, and the average efficiency is the

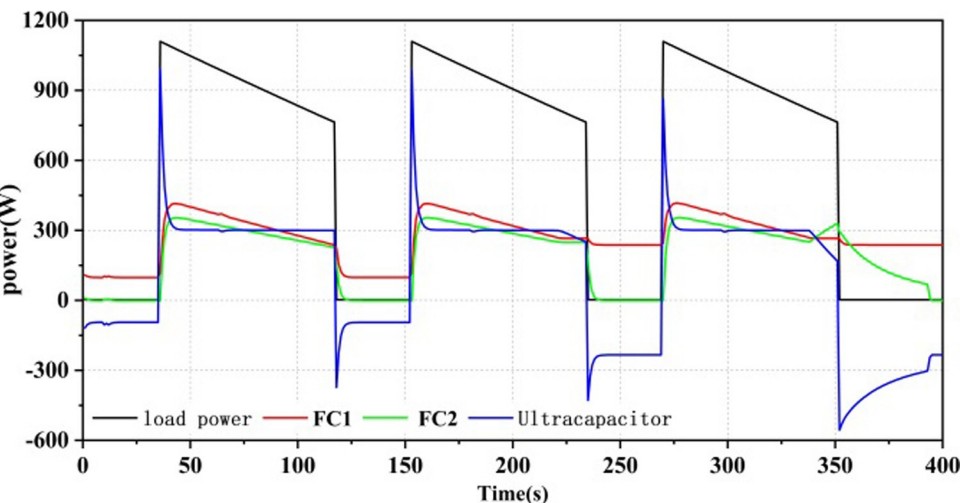

**Fig 14. A state machine strategy based on minimisation of equivalent hydrogen consumption.**

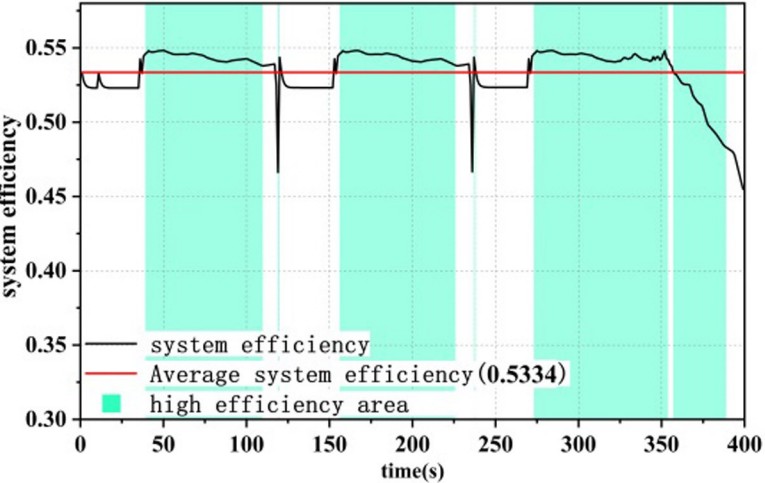

**Fig 15. Equivalent hydrogen consumption system efficiency.**

highest at 53.83%. This indicates that the proposed optimization scheme can enable the dual-stack fuel cell system to operate as much as possible within the high-efficiency operating region and improve the performance of the fuel cell power loss optimization system.

## 4.3 Comparison of hydrogen consumption and SOC

The results for the supercapacitor state of charge (SOC) and hydrogen consumption are illustrated in Figs 18 and 19, respectively. The initial SOC of the supercapacitor was set at 86%. During the operation, the SMC energy management strategy primarily considers the SOC status. At the beginning of the system operation, the SOC of the supercapacitor decreases the fastest, but the hydrogen consumption by the fuel cell is at its lowest. As the system power operation continues, the working time of the supercapacitor decreases, and the fuel cell becomes the primary power source, leading to an increase in hydrogen consumption.

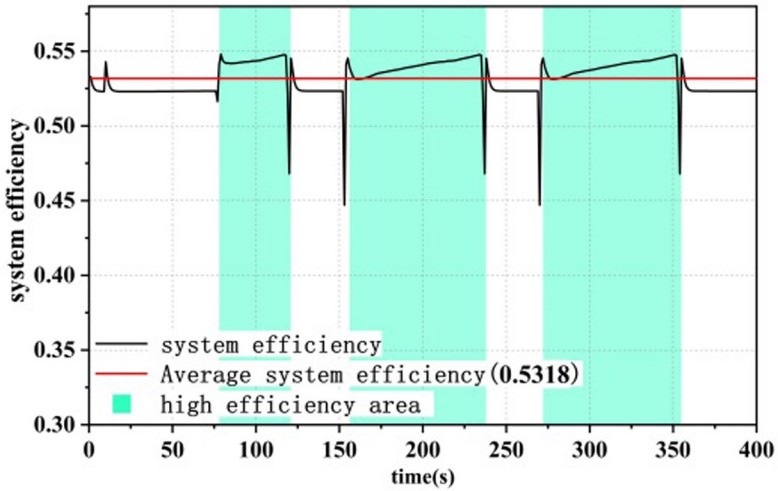

**Fig 16. State machine control system efficiency.**

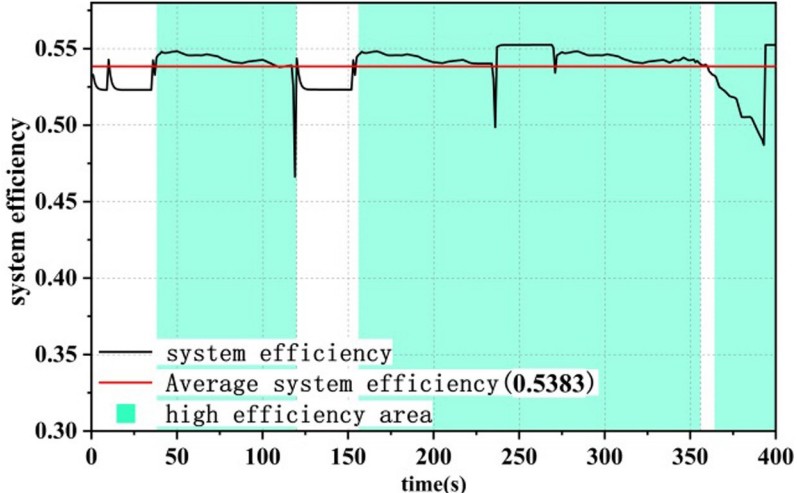

**Fig 17. A state machine strategy based on minimisation of equivalent hydrogen consumption.**

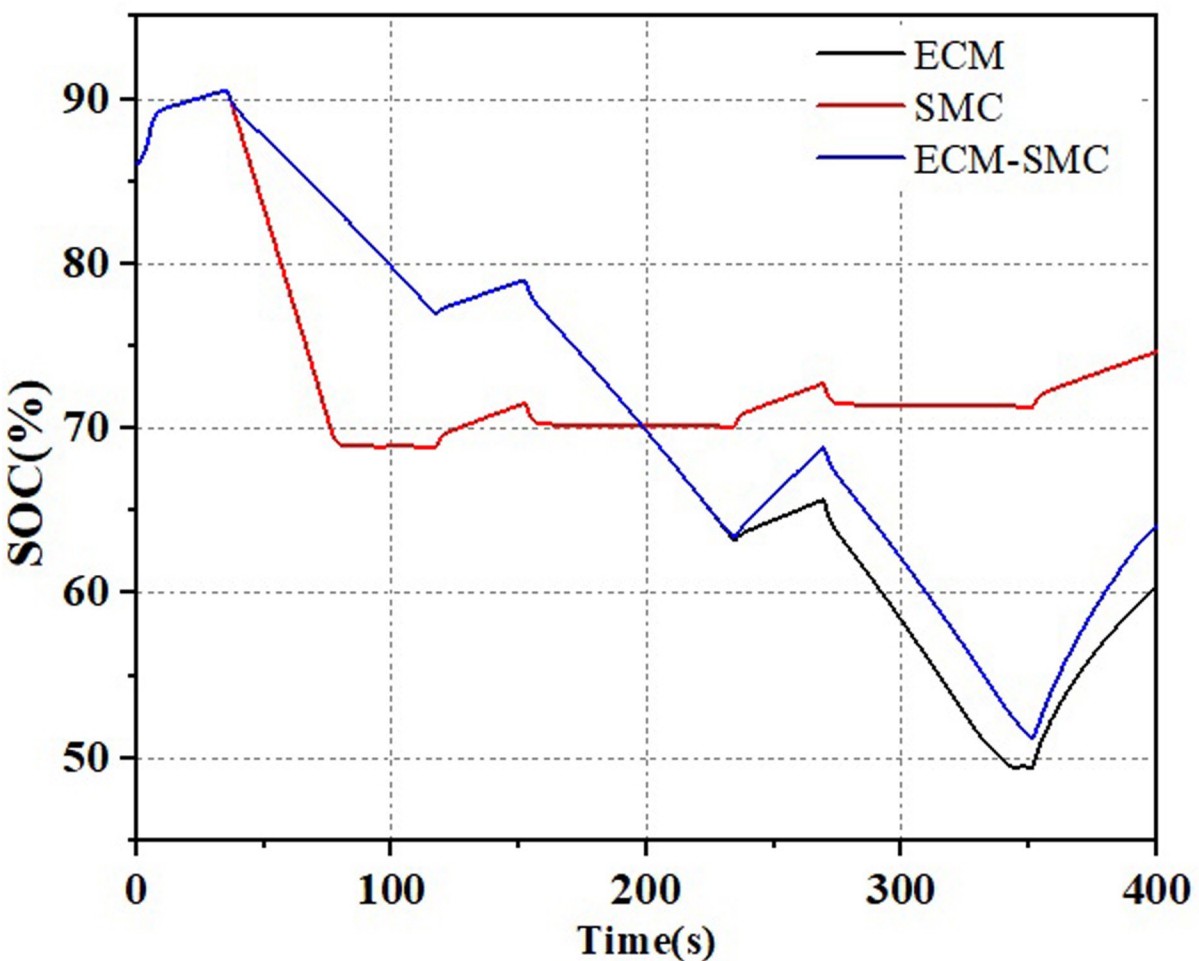

**Fig 18. Change curve of supercapacitor' SOC.**

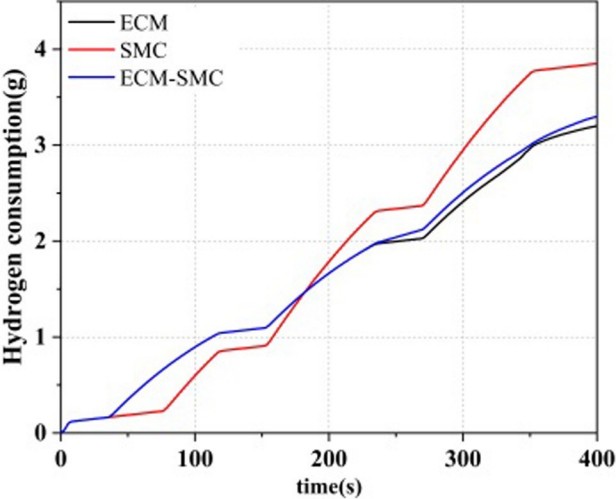

**Fig 19. Hydrogen consumption curve of the system.**

Compared to the ECM strategy, the ECM-SMC power allocation strategy exhibits a similar hydrogen consumption pattern, with an increase in the fuel cell's charging of the supercapacitor. This results in a rise in the SOC of the supercapacitor.

## 4.4 Analysis of operating stresses in fuel cell subsystems and statistical distribution of operational points

Frequent load fluctuations can elevate the operational pressure of fuel cells, consequently impacting their efficiency [17]. This is because fuel cells may experience increased resistance to electron and proton transfer when operating under high pressure, resulting in augmented energy losses, thereby influencing both the efficiency and lifespan of fuel cells. To quantify the operational stress of fuel cells and its impact on their service life, there are currently two commonly used methods. The first method involves decomposing the fuel cell load signal through

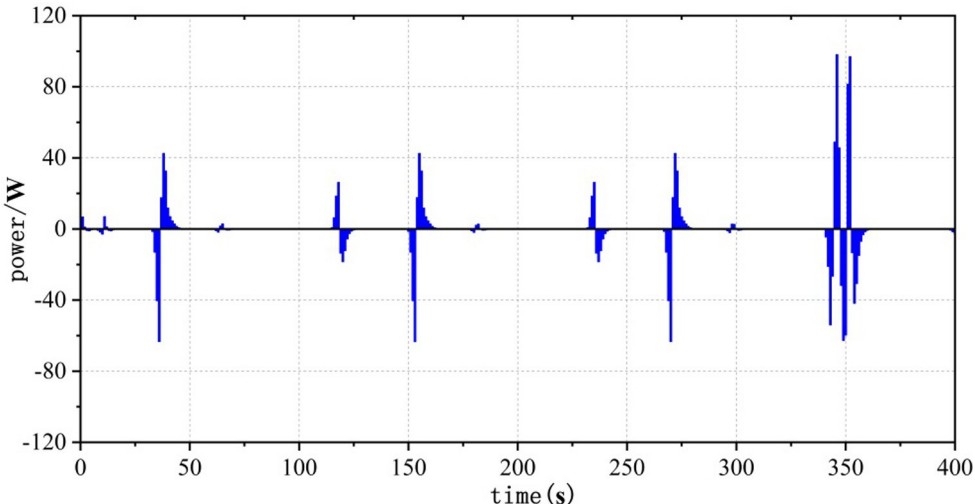

**Fig 20. ECMs Operating pressure analysis($\sigma = 6.88$).**

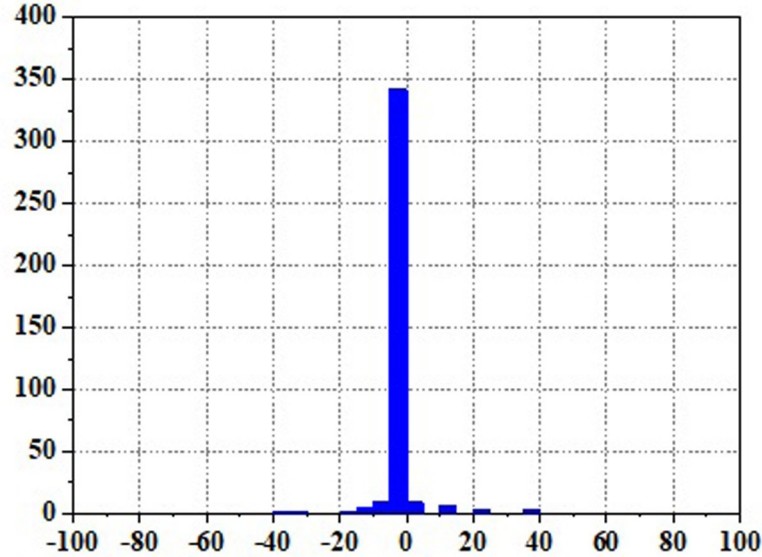

**Fig 21. ECMs Operating pressure analysis results (σ = 6.88).**

the Haar wavelet transformation, separating the high-frequency load power, and subjecting the fuel cell to stress analysis. The standard deviation of the high-frequency component σ reflects the frequency at which the fuel cell system is requested, with smaller values indicating a more stable fuel cell output power and smaller system power fluctuations and operational stresses [18]. The second method involves statistical power fluctuations and by tracking the changes in fuel cell power ΔP, the operational stress of the fuel cell can be evaluated. When the majority of the ΔP values fall into the small range near "0," it indicates low operational stress on the fuel cell.

Figs 20–31 show the operational stress analysis of the fuel cell subsystem under the three power allocation strategies. The fuel cell adaptation frequency is $[10^{-6}, 10^{-2}]$ and anything

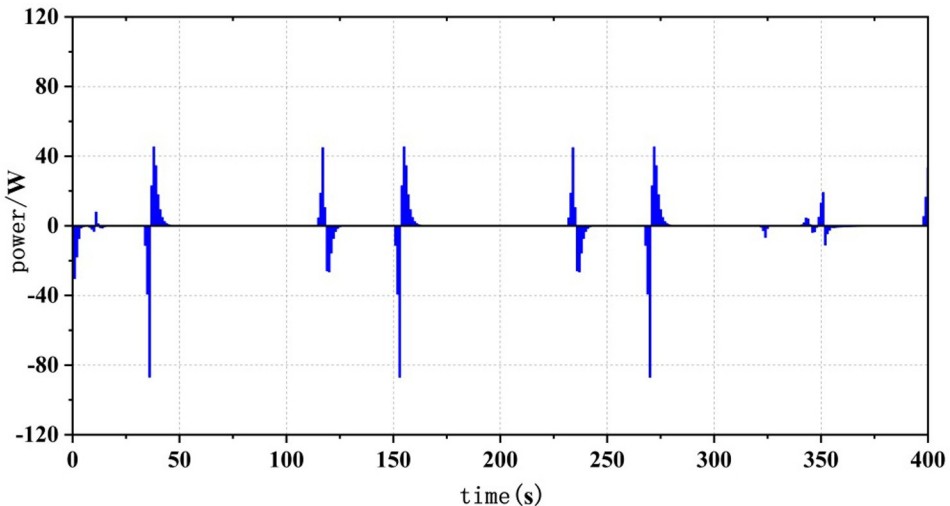

**Fig 22. ECMs Operating pressure analysis(σ = 4.27).**

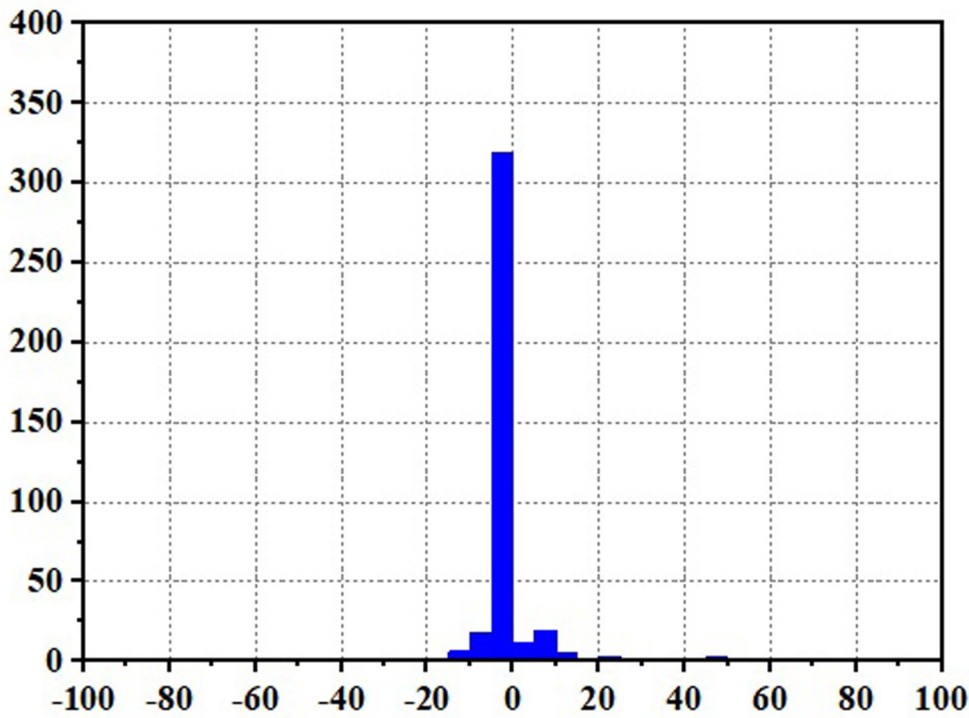

**Fig 23. ECMs Operating pressure analysis results (σ = 4.27).**

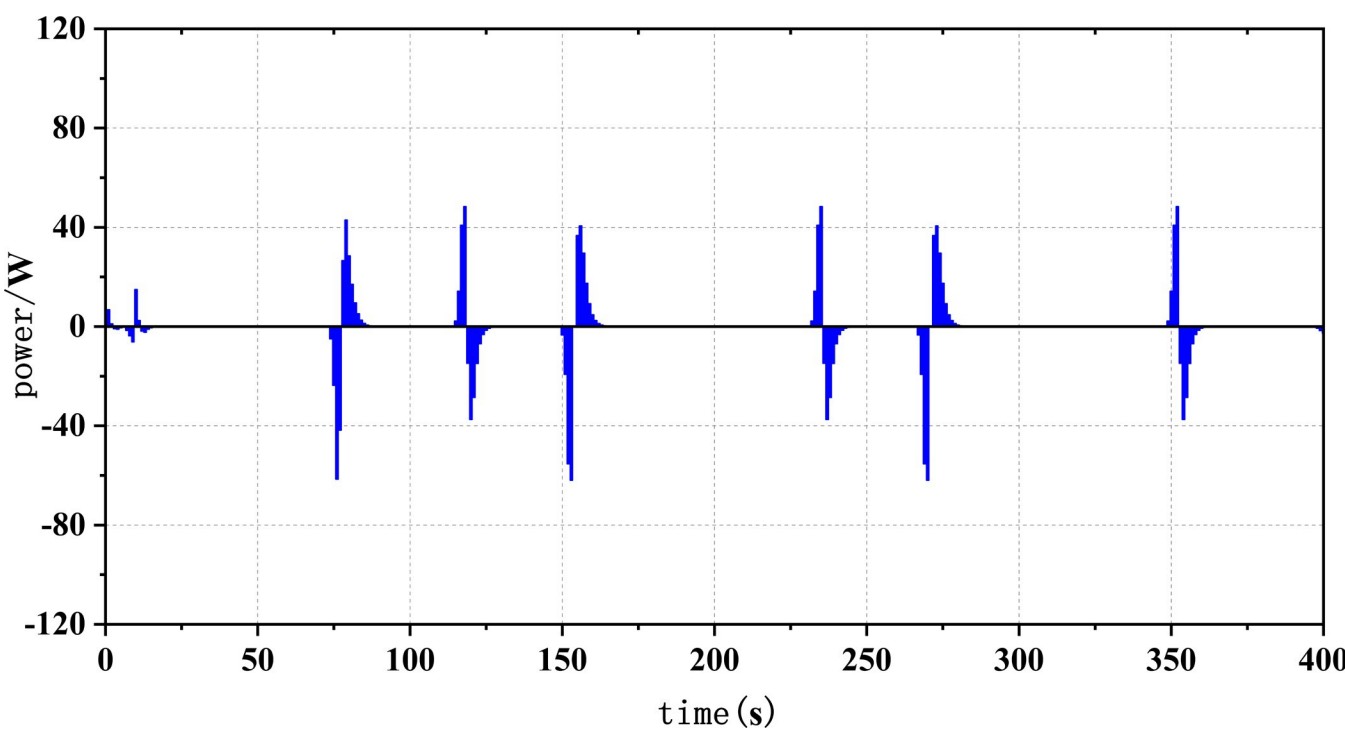

**Fig 24. SMC operating pressure analysis(σ = 5.60).**

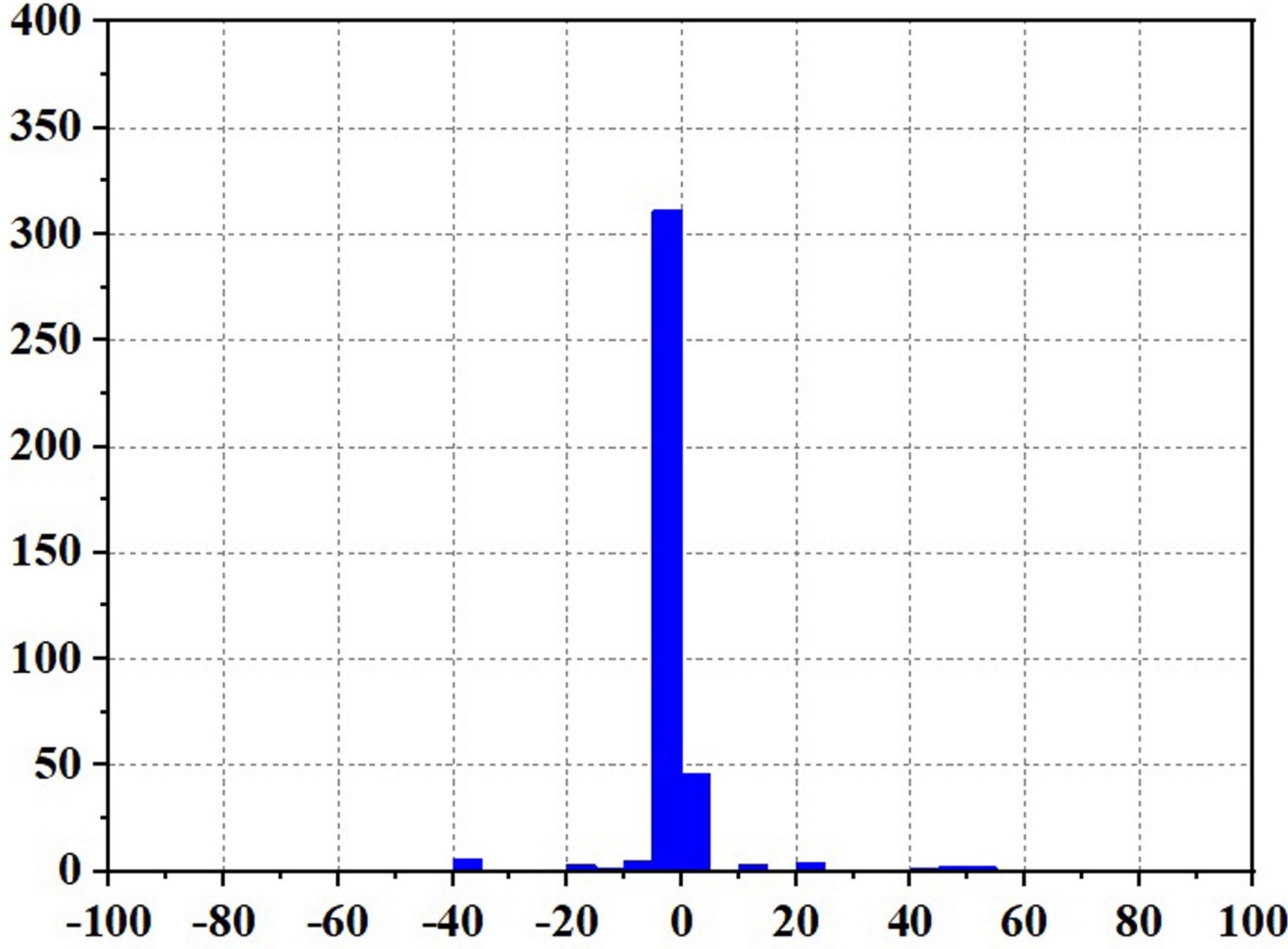

**Fig 25. SMC operating pressure analysis results (σ = 5.60).**

beyond this range is considered a high-frequency part. The operational stress is highest under the state machine-based power allocation strategy because the upper and lower bounds of fuel cell power variations were not considered during the operation process. The operational stress of the PEMFC subsystem under the ECM and ECM-SMC strategies is similar, with PEMFC1 being lower in ECMs than in ECM-SMC, while the operational stress of PEMFC2 is slightly higher. This indicates that in the power allocation strategy of ECM-SMC, PEMFC1 bears more operational stress.

In Fig 32, the distribution of operational points for the MFC system in three different scenarios is depicted. All three scenarios ensure that the MFC system operates primarily within the high-efficiency region. In the initial phase of system operation, the SMC system relies on the supercapacitor to handle a portion of the output, with the highest proportion occurring in the non-efficient region. To further analyze the operation of the MFC system and its subsystems within the high-efficiency region, the operational points within this region were statistically analyzed and converted to percentages, as shown in Fig 33. Overall, the proposed approach in this paper enhances the operational lifespan of the MFC system and its subsystems within the high-efficiency region. The MFC system accounts for 77% of the total operational

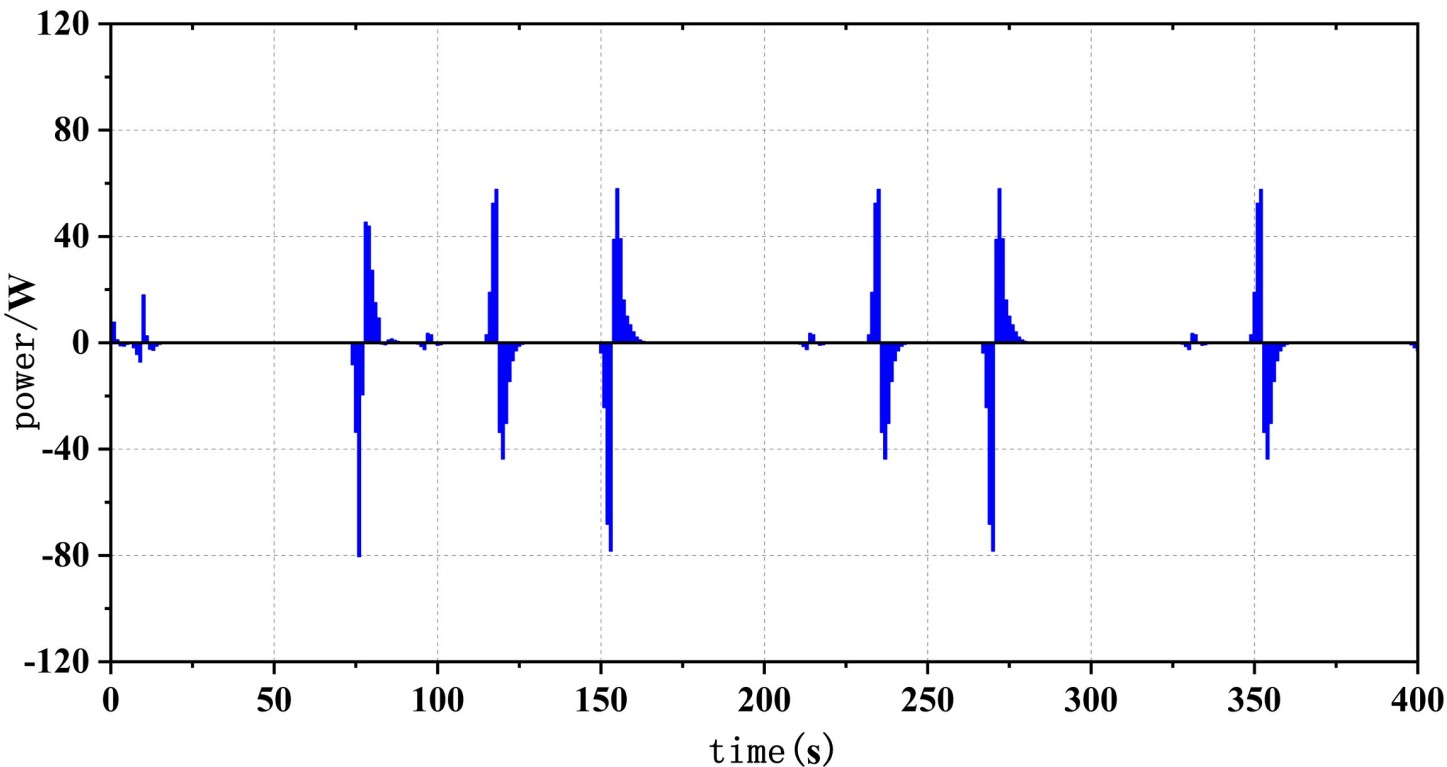

**Fig 26. SMC operating pressure analysis(σ = 7.28).**

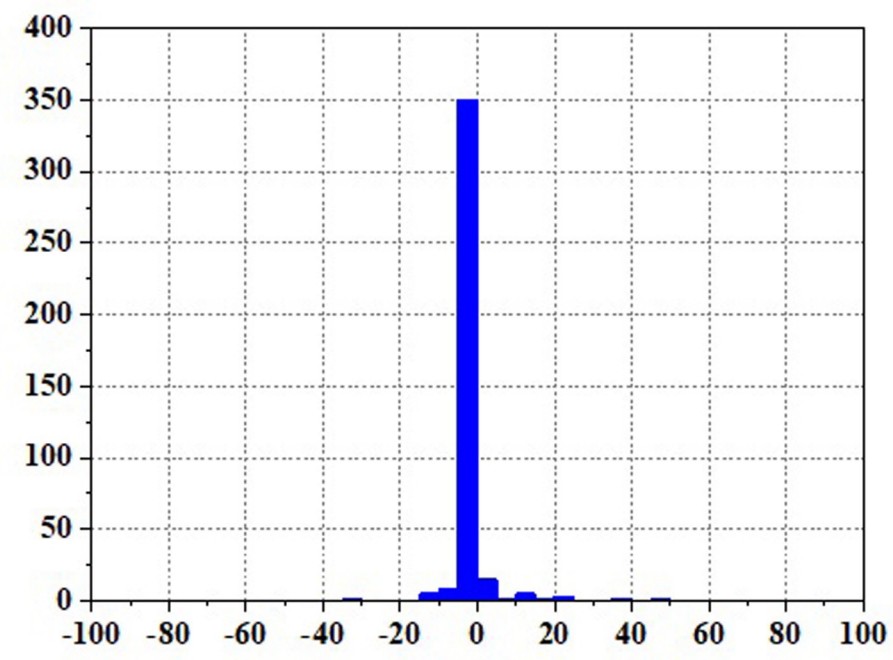

**Fig 27. SMC operating pressure analysis results (σ = 7.28).**

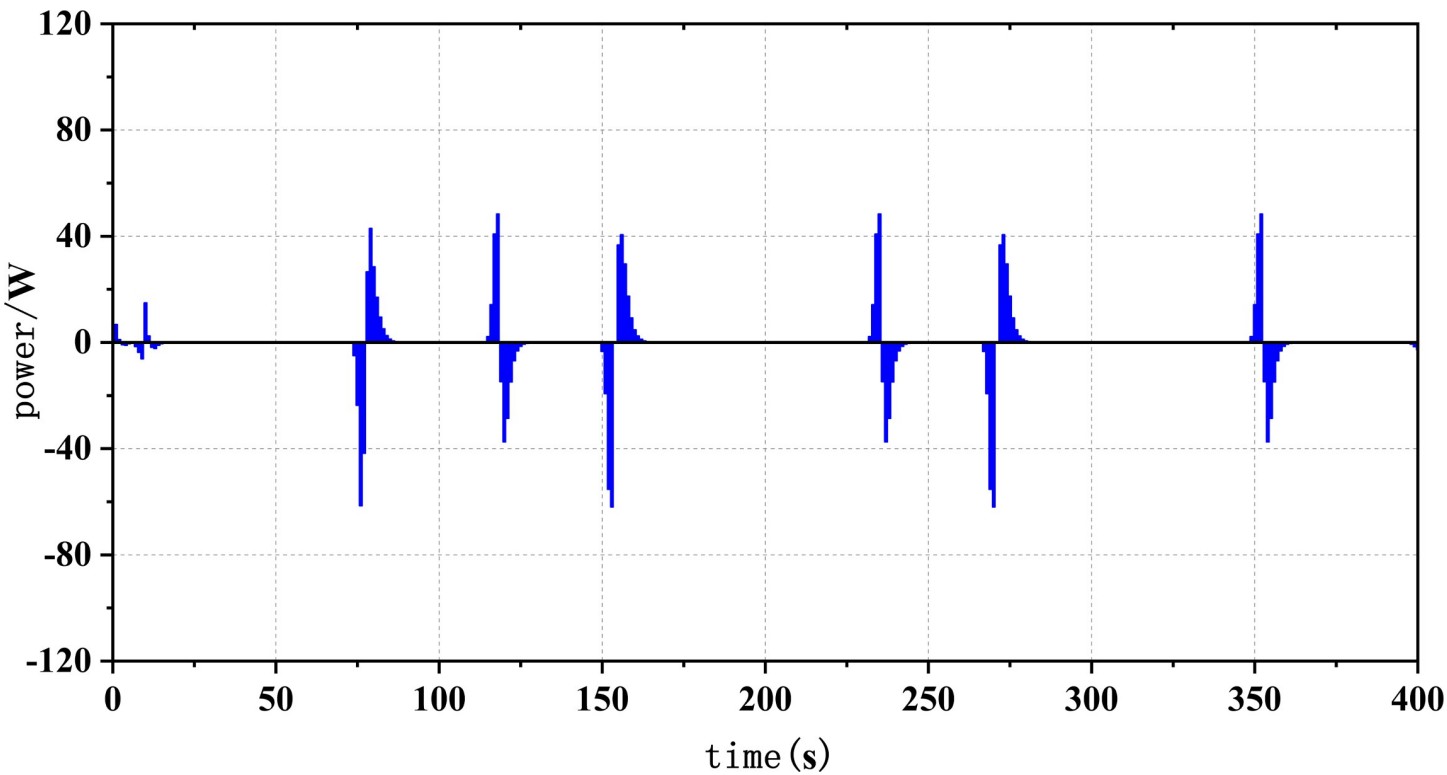

**Fig 28. SMC operating pressure analysis(σ = 6.88).**

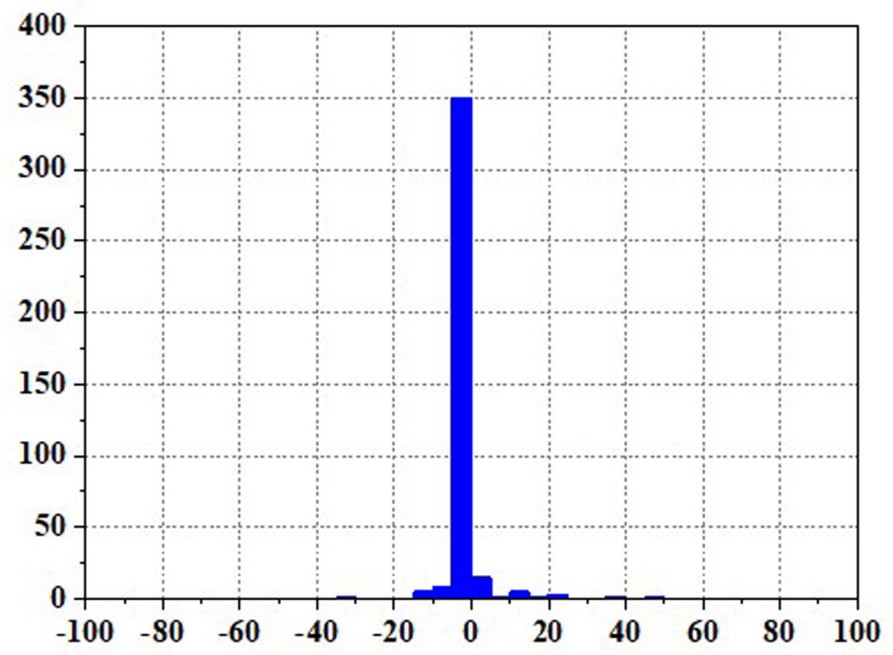

**Fig 29. SMC operating pressure analysis results (σ = 6.88).**

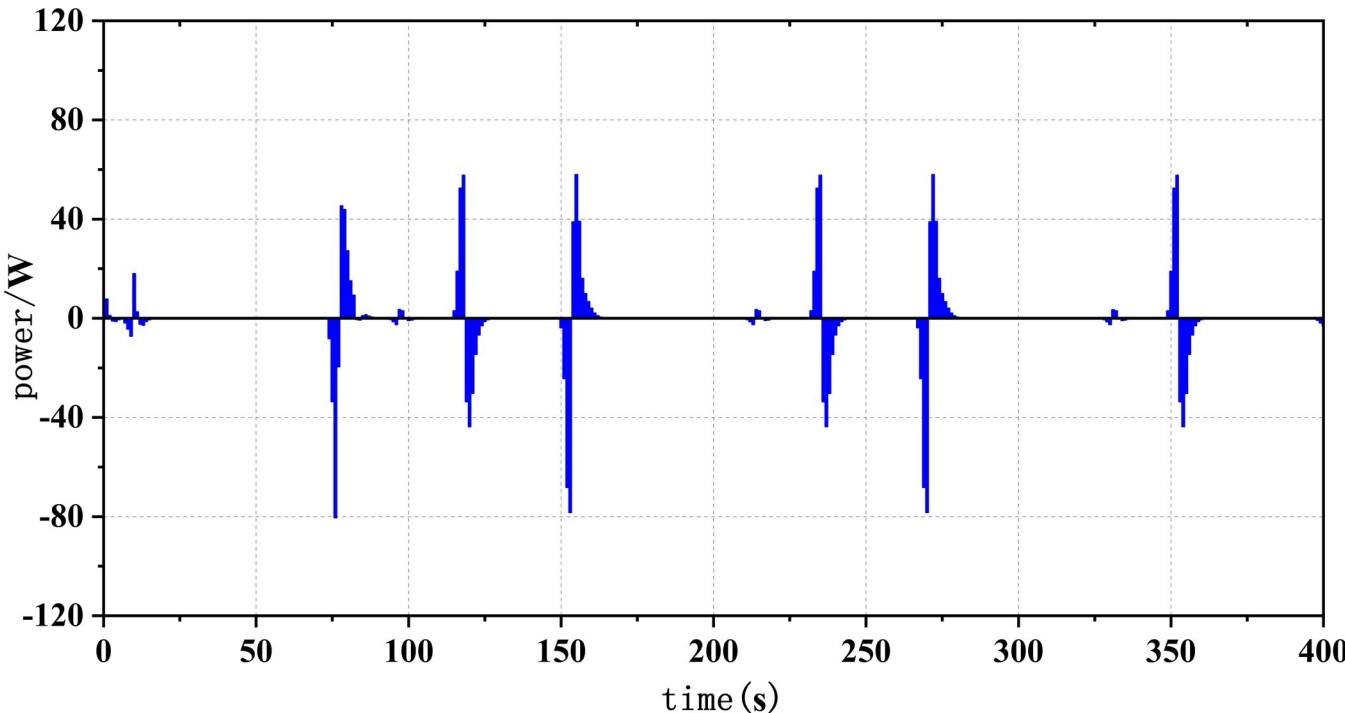

**Fig 30. SMC operating pressure analysis(σ = 4.27).**

time (12% higher than ECM, and 24% higher than SMC). Subsystem FC1 and FC2 have operational periods of 82% (21% higher than ECM, and 30% higher than SMC) and 58% (1% higher than ECM, and 7% higher than SMC), respectively.

Through a comparative analysis of the performance indicators for each scenario, the ECM-SMC method employed in this paper effectively reduces the working pressure of the PEMFC system. This reduction is beneficial in minimizing the disturbance in the output power of the PEMFC system and ultimately leads to improved system performance, as indicated in Table 3.

## 5 Conclusion

Propose an energy management strategy for a dual-stack fuel cell/supercapacitor hybrid system that combines Equivalent Consumption Minimization (ECM) and a state machine approach. This strategy is based on the real-time power allocation policy for multi-stack fuel cell systems while considering the maximum efficiency of the fuel cell system. Reverse verification reveals that in full startup mode, the maximum efficiency of the dual-stack fuel cell system is determined by the poorest-performing fuel cell system. By reverse deducing from the least efficient fuel cell system, the optimal efficiency point for the dual-stack fuel cell system and its corresponding power level can be determined.

Aim to develop an energy management strategy for a multi-source hybrid system with the goal of operating the system with lower energy consumption while increasing the operating time of the fuel cell in its high-efficiency range. This strategy calculates the optimal output power of supercapacitors at the current State of Charge (SOC) in real-time and then controls the output power of the fuel cell system using a designed state machine. Simulation results indicate that this method provides more stable power fluctuations, a slight increase in

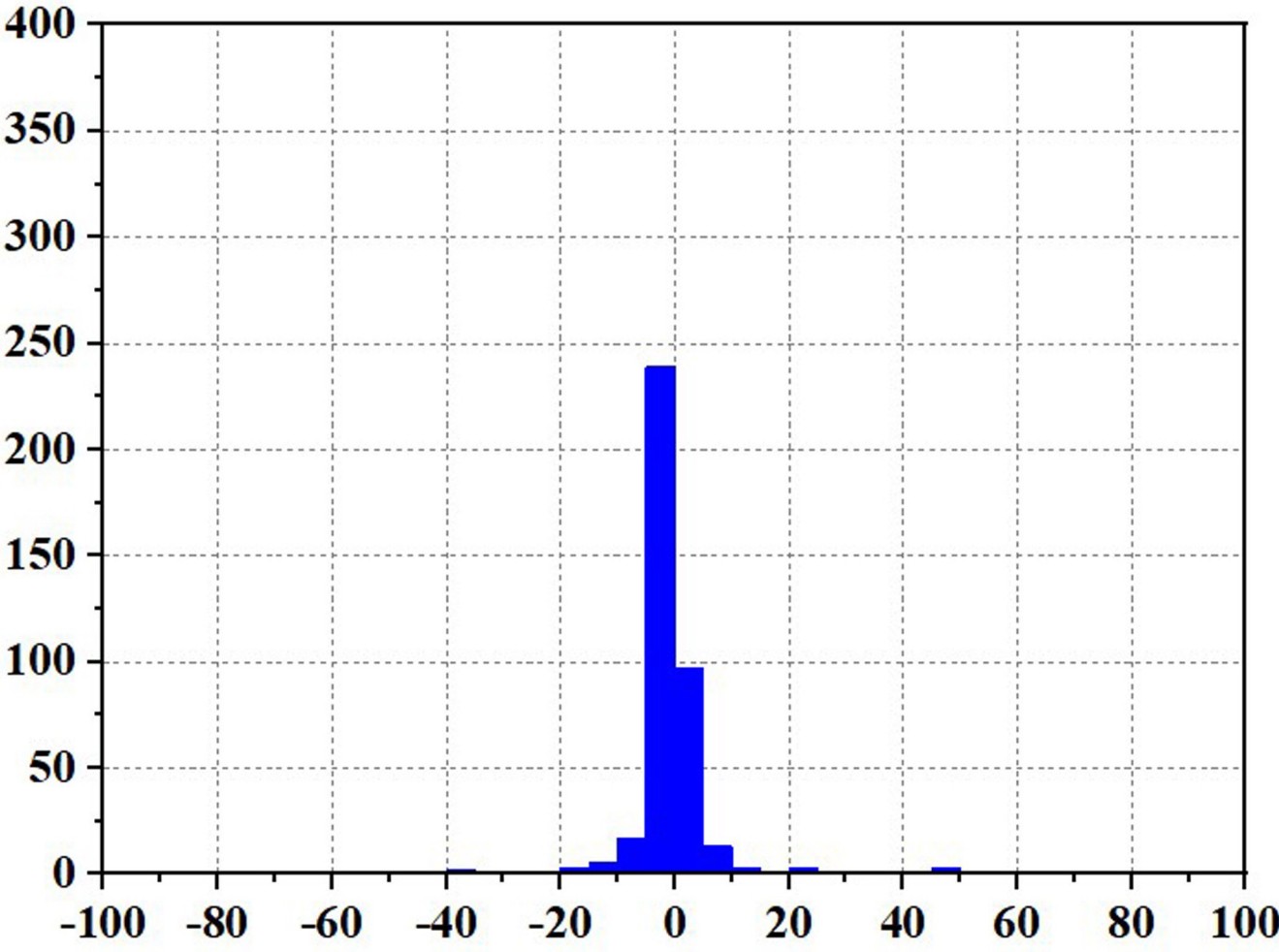

**Fig 31. SMC operating pressure analysis results (σ = 4.27).**

hydrogen consumption compared to the ECM strategy, but an improvement in SOC. Furthermore, it results in lower hydrogen consumption compared to the State Machine Control (SMC) strategy, increased average efficiency, and a higher proportion of operation in the high-efficiency zone for the fuel cell system and its subsystems. This approach not only effectively identifies the high-efficiency zone of fuel cells but also enhances the system's high-efficiency operating range, reduces hydrogen consumption, and improves system stability.

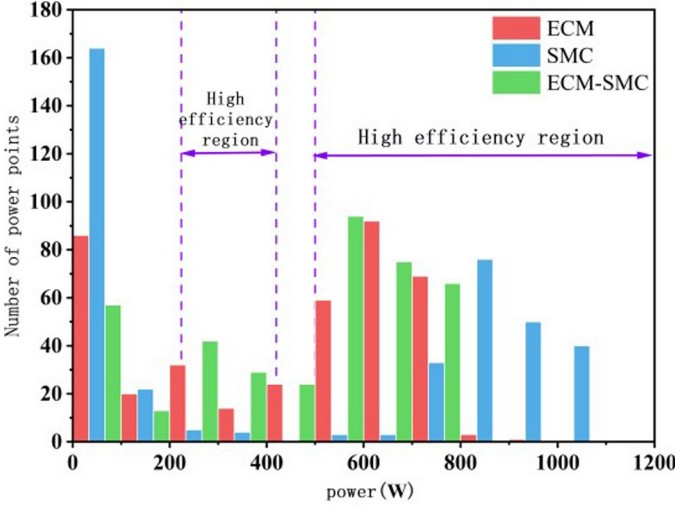

**Fig 32. Distribution of MFC operating points.**

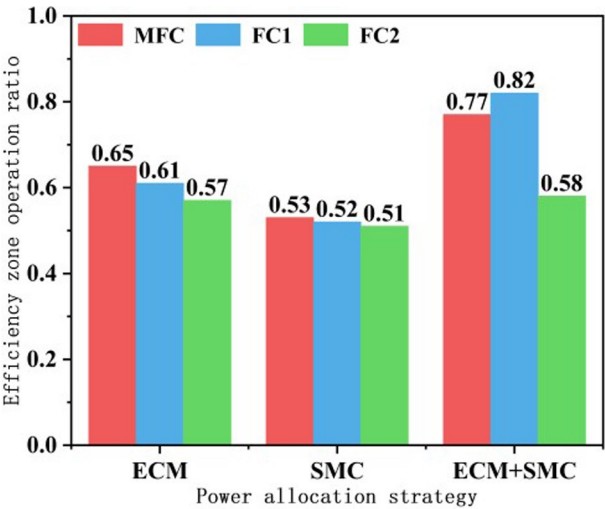

**Fig 33. Three methods of efficient zone occupancy.**

**Table 3. Comparative analysis of performance of different control strategies.**

|  | ECM | SMC | ECM+SMC |
|---|---|---|---|
| Mean efficiency of MFC (%) | 53.34 | 53.18 | 53.83 |
| Hydrogen consumption | 3.20 | 3.849 | 3.296 |
| SOC variation of the supercapacitor (%) | 86.16–60.39 | 86.16–74.67 | 86.16–64.14 |
| Proportion of high-efficiency region of MFC (%) | 65.25 | 53.00 | 77.50 |
| Proportion of high-efficiency region of FC1 (%) | 61.25 | 52.00 | 82.00 |
| Proportion of high-efficiency region of FC2 (%) | 57.75 | 51.25 | 58.75 |
| Operational stress analysis of FC1 | 6.881 | 5.604 | 6.881 |
| Operational stress analysis of FC2 | 4.273 | 7.289 | 4.273 |

## Supporting information

**S1 Data.**
(XLSX)

## Author Contributions

**Conceptualization:** Jianfeng Zhao.

**Data curation:** Jianfeng Zhao, Yan Qin.

**Formal analysis:** Mengjie Li.

**Methodology:** Mengjie Li, Qianchao Liang.

**Project administration:** Yongbao Liu.

**Resources:** Yongbao Liu.

**Software:** Mengjie Li.

**Validation:** Mengjie Li.

**Visualization:** Mengjie Li.

**Writing – original draft:** Mengjie Li.

**Writing – review & editing:** Mengjie Li.

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
