## [Decision Letter · Decision Letter 0]

18 Mar 2024

PONE-D-24-04478Research on State Machine Control Optimization of Double-stack Fuel Cell/Super Capacitor Hybrid SystemPLOS ONE

Dear Dr. Li,

Thank you for submitting your manuscript to PLOS ONE. After careful consideration, we feel that it has merit but does not fully meet PLOS ONE’s publication criteria as it currently stands. Therefore, we invite you to submit a revised version of the manuscript that addresses the points raised during the review process.

We look forward to receiving your revised manuscript.

Kind regards,

Jim P. Zheng

Academic Editor

PLOS ONE

A clean copy of the edited manuscript (uploaded as the new *manuscript* file).

5. PLOS requires an ORCID iD for the corresponding author in Editorial Manager on papers submitted after December 6th, 2016. Please ensure that you have an ORCID iD and that it is validated in Editorial Manager. To do this, go to ‘Update my Information’ (in the upper left-hand corner of the main menu), and click on the Fetch/Validate link next to the ORCID field. This will take you to the ORCID site and allow you to create a new iD or authenticate a pre-existing iD in Editorial Manager. Please see the following video for instructions on linking an ORCID iD to your Editorial Manager account: https://www.youtube.com/watch?v=_xcclfuvtxQ.

Reviewers' comments:

Reviewer's Responses to Questions

**Comments to the Author**

1. Is the manuscript technically sound, and do the data support the conclusions?

Reviewer #1: Yes

2. Has the statistical analysis been performed appropriately and rigorously? 

Reviewer #1: N/A

3. Have the authors made all data underlying the findings in their manuscript fully available?

Reviewer #1: No

4. Is the manuscript presented in an intelligible fashion and written in standard English?

Reviewer #1: Yes

5. Review Comments to the Author

Reviewer #1: Good paper but requires the following revisions:

1. Abstract: should have a clear statement of the paper main findings and its significance.

2. Introduction: rephrase te following statement to ensure its clarity and brevity.

With the development of fuel cell systems towards high power, high power fuel cells in addition to the final installation problems, due to the large size, there may be mechanical failure, resulting in gas leakage; It may also lead to gas non-uniformity in the internal reaction of the fuel cell [1], resulting in poor voltage consistency of each single cell in the stack, increasing the difficulty of system operation.

3. Introduction: proofread the paper before submission for review. For example, see the following unnecessary repetition:

the train can reach a maximum speed of 140km per hour, a maximum speed of 140km per hour.

4. Paper should have an abbreviation list, or all symbols should be identified when first presented.

5. Page 10: Add a reference after the following statement:

Feroldi proposed using a state machine policy to ensure that the fuel cell in the hybrid energy system operates in the high-efficiency region, as shown in Figure 2.

6. Page 10: Give brief explanation of the multiple peaks in relation to inflection point.

The efficiency chart of multi-stack fuel cell systems exhibits multiple peaks due to the presence of the inflection point.

7. Page 11: repetition of a word:

The strategy architecture diagram is shown in Figure Figure 5,

8. Table1: last row is bold; is there a significance for that or just typo.

9. Table 2: column title should be in English

10. Figure 12 a, b, c: explain the meaning and significance of the spikes.

11. Section 3.4/paragraph one: explain the relationship between stress and delta P.

When the majority of the ΔP values fall into the small range near "0," it indicates low operational stress on the fuel cell.

12. Table 3: correct the table title, it is table 3 and not fig 3. In addition, add units or % to hydrogen consumption and operational stress analysis.

13. Conclusion: rewrite the first sentence. Maybe you can start by saying the paper propose…. Or in these investigations it is proposed….

Propose an energy management strategy for a dual-stack fuel cell/supercapacitor hybrid system that combines Equivalent Consumption Minimization (ECM) and a state machine approach.

14. Conclusion: Why the poorest performing cell? Clarify in the previous sections and briefly state in the conclusion.

Reverse verification reveals that in full startup mode, the maximum efficiency of the dual-stack fuel cell system is determined by the poorest-performing fuel cell system.

15. Conclusion: second paragraph requires writing. You may state … the investigation aimed to….

Aim to develop an energy management strategy for a multi-source hybrid system with the goal of operating the system with lower energy consumption while increasing the operating time of the fuel cell in its high-efficiency range

6. PLOS authors have the option to publish the peer review history of their article (what does this mean?). If published, this will include your full peer review and any attached files.

Reviewer #1: No

---

## [Author Response · Author response to Decision Letter 0]

16 May 2024

Thank you very much for the expert's advice. I have revised and explained the expert's suggestions one by one. And submitted it to the file. Please review.

1. Abstract: should have a clear statement of the paper main findings and its significance.(modified)

2. Introduction: rephrase te following statement to ensure its clarity and brevity.(modified)

With the development of fuel cell systems towards high power, high power fuel cells in addition to the final installation problems, due to the large size, there may be mechanical failure, resulting in gas leakage; It may also lead to gas non-uniformity in the internal reaction of the fuel cell [1], resulting in poor voltage consistency of each single cell in the stack, increasing the difficulty of system operation.

3. Introduction: proofread the paper before submission for review. For example, see the following unnecessary repetition:(modified)

the train can reach a maximum speed of 140km per hour, a maximum speed of 140km per hour.

4. Paper should have an abbreviation list, or all symbols should be identified when first presented.(modified)

5. Page 10: Add a reference after the following statement:

Feroldi proposed using a state machine policy to ensure that the fuel cell in the hybrid energy system operates in the high-efficiency region, as shown in Figure 2.(modified)

6. Page 10: Give brief explanation of the multiple peaks in relation to inflection point.(modified)

The efficiency chart of multi-stack fuel cell systems exhibits multiple peaks due to the presence of the inflection point.

7. Page 11: repetition of a word:(modified)

The strategy architecture diagram is shown in Figure Figure 5,

8. Table1: last row is bold; is there a significance for that or just typo.(modified)

9. Table 2: column title should be in English(modified)

10. Figure 12 a, b, c: explain the meaning and significance of the spikes.(modified)

The efficiency of this figure is simulated according to the load situation in Figure 11. Each peak out is caused by the sudden load drop in the power of the fuel cell, and its efficiency will naturally have a mutation.

11.Section 3.4/paragraph one: explain the relations.(modified)

The study of the influence of frequent load changes on fuel cells is closely related to the efficiency study of fuel cells. The efficiency of a fuel cell refers to its ability to convert chemical energy into electricity, which is affected by a variety of factors, including operating pressure and load changes, etc.

Frequent load changes can increase the operating pressure of fuel cells, thus affecting their efficiency. Because when the fuel cell operates at high pressure, there may be increased resistance to electron transmission and proton transmission, resulting in increased energy loss, which then affects the efficiency of the fuel cell.

Therefore, the study of fuel cell efficiency needs to consider the influence of load change on the operating pressure of fuel cells. By analyzing the pressure change of fuel cell and further studying how to optimize the fuel cell system to improve its stability and efficiency, it can provide an important reference for improving the efficiency of fuel cell.

---

## [Editor Report · Decision Letter 1]

29 May 2024

Research on State Machine Control Optimization of Double-stack Fuel Cell/Super Capacitor Hybrid System

PONE-D-24-04478R1

Dear Dr. Li,

We’re pleased to inform you that your manuscript has been judged scientifically suitable for publication and will be formally accepted for publication once it meets all outstanding technical requirements.

Kind regards,

Jim P. Zheng

Academic Editor

PLOS ONE
---

## [Editor Report · Acceptance letter]

20 Jun 2024

PONE-D-24-04478R1 

PLOS ONE

Dear Dr. Li, 

I'm pleased to inform you that your manuscript has been deemed suitable for publication in PLOS ONE. Congratulations! Your manuscript is now being handed over to our production team.

Kind regards, 

on behalf of

Dr. Jim P. Zheng 

Academic Editor

PLOS ONE